# Synaptic dysfunction induced by glycine-alanine dipeptides in C9orf72-ALS/FTD is rescued by SV2 replenishment

Brigid K Jensen[1] (ID), Martin H Schuldi[2], Kevin McAvoy[1], Katelyn A Russell[1], Ashley Boehringer[1], Bridget M Curran[3], Karthik Krishnamurthy[1], Xinmei Wen[1], Thomas Westergard[1], Le Ma[3], Aaron R Haeusler[1] (ID), Dieter Edbauer[2] (ID), Piera Pasinelli[1] & Davide Trotti[1,*] (ID)

## Abstract

The most common cause of amyotrophic lateral sclerosis (ALS) and frontotemporal dementia (FTD) is an intronic hexanucleotide repeat expansion in the *C9orf72* gene. In disease, RNA transcripts containing this expanded region undergo repeat-associated non-AUG translation to produce dipeptide repeat proteins (DPRs), which are detected in brain and spinal cord of patients and are neurotoxic both *in vitro* and *in vivo* paradigms. We reveal here a novel pathogenic mechanism for the most abundantly detected DPR in ALS/FTD autopsy tissues, poly-glycine-alanine (GA). Previously, we showed motor dysfunction in a GA mouse model without loss of motor neurons. Here, we demonstrate that mobile GA aggregates are present within neurites, evoke a reduction in synaptic vesicle-associated protein 2 (SV2), and alter $Ca^{2+}$ influx and synaptic vesicle release. These phenotypes could be corrected by restoring SV2 levels. In GA mice, loss of SV2 was observed without reduction of motor neuron number. Notably, reduction in SV2 was seen in cortical and motor neurons derived from patient induced pluripotent stem cell lines, suggesting synaptic alterations also occur in patients.

**Keywords** amyotrophic lateral sclerosis; C9orf72; dipeptide repeat proteins; motor deficit; synaptic transmission
**Subject Categories** Genetics, Gene Therapy & Genetic Disease; Neuroscience; Stem Cells & Regenerative Medicine

## Introduction

An intronic hexanucleotide repeat expansion (GGGGCC)$_n$ in the C9orf72 gene is the most frequent genetic cause of amyotrophic lateral sclerosis (ALS) and frontotemporal dementia (FTD) (Couthouis *et al*, 2011; DeJesus-Hernandez *et al*, 2011; Renton *et al*, 2011). Evidence has been produced in support of three pathogenic mechanisms linked to this disease-causing expansion: (i) a loss-of-function mechanism due to decreased levels of C9orf72 mRNA expression (Ciura *et al*, 2013; Therrien *et al*, 2013; Shi *et al*, 2018); (ii) RNA transcripts generated from the expanded region gain toxic functions either by forming RNA–DNA hybrids that inhibit transcription and/or by sequestering RNA-binding proteins in nuclear foci (Gendron *et al*, 2013; Gitler & Tsuiji, 2016); (iii) through repeat-associated non-AUG translation, expanded RNA transcripts encode six dipeptide repeat (DPR) proteins (sense: poly-glycine-alanine or GA, poly-glycine-arginine or GR, poly-glycine-proline or GP; antisense: poly-proline-arginine or PR, poly-proline-alanine or PA, and poly-proline-glycine or PG), which display different toxic profiles in different organismal and cell culture systems (Ash *et al*, 2013; Mori *et al*, 2013; Zu *et al*, 2013; Mizielinska *et al*, 2014; Haeusler *et al*, 2016; Wen *et al*, 2017). Each of these mechanisms invokes widespread cellular responses and, in some cases, culminates in neuronal toxicity and cognitive and motor dysfunctions resembling the human disease phenotype (Kwon *et al*, 2014; Mizielinska *et al*, 2014; Wen *et al*, 2014, 2017; Tao *et al*, 2015).

Evidence is mounting for dysfunctional neuromuscular junction (NMJ) transmission as an early event in C9orf72-ALS (Sareen *et al*, 2013; Devlin *et al*, 2015). Notably, two independent studies in *Drosophila melanogaster* larvae have investigated structural attributes of the NMJ in animals expressing GGGGCC repeats. In these studies, both groups reported drastic reductions in the number of NMJ active zones in larvae expressing 30 or more GGGGCC repeats (Freibaum *et al*, 2015; Zhang *et al*, 2015). Detailed analysis also revealed a significant decrease in synaptic bouton number (Freibaum *et al*, 2015), as well as reduced synaptic quantal content leading to attenuated evoked potentials (Zhang *et al*, 2015). However, it is not clear whether these effects are due to toxic gain of function of expanded RNA transcripts, DPRs, or a combination of both acting

1   Jefferson Weinberg ALS Center, Department of Neuroscience, Vickie and Jack Farber Institute for Neuroscience, Thomas Jefferson University, Philadelphia, PA, USA
2   German Center for Neurodegenerative Diseases (DZNE), Munich, Germany
3   Department of Neuroscience, Vickie and Jack Farber Institute for Neuroscience, Thomas Jefferson University, Philadelphia, PA, USA
    *Corresponding author. Tel: +1 215 955 8416; E-mail: davide.trotti@jefferson.edu

synergistically. Based on numerous recent reports investigating these potential pathogenic mechanisms both *in vitro* and in animal models, a substantial focus has now been placed on better investigating the effects of DPRs and their potential pathogenic consequences (Gitler & Tsuiji, 2016; Freibaum & Taylor, 2017; Wen *et al*, 2017). As such, herein we have chosen to limit our mechanistic analysis regarding C9orf72 repeat expansion effects on synaptic function to the contribution of DPR-mediated pathological changes, in the absence of potential confounds due to the concomitant presence of GGGGCC repeat-containing RNA transcripts.

Abundant evidence for *in vivo* and *in vitro* effects of DPRs has amassed, pointing to pathogenic roles for these dipeptides (Kwon *et al*, 2014; Mizielinska *et al*, 2014; Wen *et al*, 2014; Tao *et al*, 2015; Boeynaems *et al*, 2016; Schludi *et al*, 2017). In fly models as well as primary neurons, arginine-containing DPRs (GR and PR) evoke the most robust toxicity (Mizielinska *et al*, 2014; Wen *et al*, 2014; Boeynaems *et al*, 2016). GR and PR both form aggregates in the nucleus and result in a host of cellular abnormalities including nuclear transport defects, protein mislocalization, altered RNA processing, and nucleolar stress (Kwon *et al*, 2014; Tao *et al*, 2015; Kim & Taylor, 2017). While not as overtly toxic, GA also contributes to neuronal dysfunction through ER stress, proteasome impairment, and sequestration of Unc119, HR23, and nucleocytoplasmic transport proteins (May *et al*, 2014; Zhang *et al*, 2016). In contrast, marginal or no toxic effects have been attributed to PA and GP/PG (Zu *et al*, 2013; Wen *et al*, 2014).

While DPR inclusions have been detected throughout the CNS of C9orf72-ALS/FTD patients, the most widespread and abundantly observed are GA[+] inclusions (Ash *et al*, 2013; Gendron *et al*, 2013; Mori *et al*, 2013; Zu *et al*, 2013). In contrast to the robust toxicity resulting from expression of the arginine-rich DPRs, cellular impairments mediated by GA appear to be more subtle. Two mouse models have independently demonstrated motor deficits stemming from neuronal GA expression by 6 months of age. When confined to the cortex, $GA_{50}$ aggregates lead to neuronal loss, brain atrophy, and mild motor and cognitive behavioral deficits (Liu *et al*, 2016), whereas when expressed in spinal cord and brainstem, aggregates positive for $GA_{149}$ trigger significant motor deficits in the absence of overt neuronal loss (Schludi *et al*, 2017). From this study, our findings suggest that in motor neurons *in vivo*, GA-mediated synaptic deficits may precede cell death (Schludi *et al*, 2017). This possibility highlights the necessity for a deeper investigation of subcellular repercussions in neurons coping with expression and aggregation of GA dipeptides, in order to identify mechanisms that may be targeted in a therapeutic window before cell death.

We and others have previously demonstrated that GA expression in primary neurons results in reduced neurite outgrowth and cellular toxicity through proteasome impairment and ER stress (May *et al*, 2014; Zhang *et al*, 2014). However, the consequences of GA inclusions on neuronal signaling have not yet been examined. In the present study, we have demonstrated that GA aggregates are mobile within axonal and dendritic neuronal compartments. These GA aggregate expressing neurons display disrupted $Ca^{2+}$ influx, selective down-regulation of the synaptic vesicle-associated protein 2 (SV2), ablated synaptic release, and increased risk of death. Through immunoblot analysis of spinal cords and synaptic staining at neuromuscular junctions from $GA_{149}$ transgenic mice, which displayed abnormal gait and progressive balance impairment but no

appreciable neuronal loss, we have confirmed the selective loss of SV2 expression. Furthermore, we have extended our findings to patient-derived cells, where the full complement of C9orf72 repeat expansion pathogenic mechanisms are potentially at play. In induced pluripotent stem cells differentiated into cortical or motor neurons, neurons derived from C9orf72 repeat expansion carriers again displayed specific reduction of SV2 levels compared with control cells. These *in vivo* and patient-derived validations of our findings suggest that this molecular phenotype could underlie or contribute to the prodromal progression of disease symptoms. Most crucially, we have been able to restore synaptic function and rescue cellular toxicity through targeted upregulation of SV2 in GA-expressing primary neurons. These findings suggest that GA-mediated alterations of SV2 levels and localization are reversible processes, from which neurons can convalesce, even under the continued presence of GA aggregates.

## Results

### GA inclusions are found in axons and dendrites of neurons

Immature primary cortical neurons (DIV4) expressing $GA_{149}$-myc displayed reduced dendritic complexity and sign of apoptosis (May *et al*, 2014). Nevertheless, in an extensive GA repeat length response curve in mature primary cortical (transfected at 7 DIV) and motor neurons (transfected at 5 DIV), we did not observe significant toxicity at any GA length up to 400 repeats, at 7 or 3 days post-transfection, respectively (Wen *et al*, 2014). We have also noted in the animal model of $GA_{149}$-CFP expression that these mice developed behavioral and motor abnormalities, such as abnormal gait and progressive balance impairment in the absence of neuronal loss (Schludi *et al*, 2017), suggesting dysfunctional neuronal activity.

We and others have established that when expressed in a variety of cell lines and in primary rodent neurons, GA forms dense cytosolic aggregates (May *et al*, 2014; Wen *et al*, 2014; Zhang *et al*, 2014). Indeed even when transmitted from cell to cell, GA maintains the tendency for cytosolic accumulation and aggregation (Chang *et al*, 2016; Westergard *et al*, 2016). As post-mitotic neurons have distinctive morphology and complex axonal and dendritic arborization architectures, in our prior work we also examined GA-expressing neurons at high spatial resolution (Wen *et al*, 2014). Mature cortical and motor neurons expressing $GA_{50}$ (eGFP-tagged) were counterstained with neuronal marker SMI-32. Intriguingly, in addition to cytoplasmic aggregates in the cell soma, eGFP-GA inclusions were also present within neurites (Wen *et al*, 2014). This localization pattern was unique to the GA dipeptide as we did not observe any of the other dipeptides within neuritic processes (Wen *et al*, 2014). To begin this present work, in order to determine whether poly-GA aggregates are found in neurites over time, we performed a longitudinal assessment of aggregates, in which $GA_{50}$ was expressed in mature cortical neurons, with fixation and staining on subpopulations of cells at specific time points. Cells were counterstained with SMI-32. At all time points examined (2, 4, or 8 days of expression), distinctive eGFP[+] GA aggregates could be found in neuritic regions (Fig 1). We confirmed that even following 8 days of expression, GFP alone without associated GA peptides does not induce GFP aggregation. Diffuse, cell-filling GFP expression can still be seen in

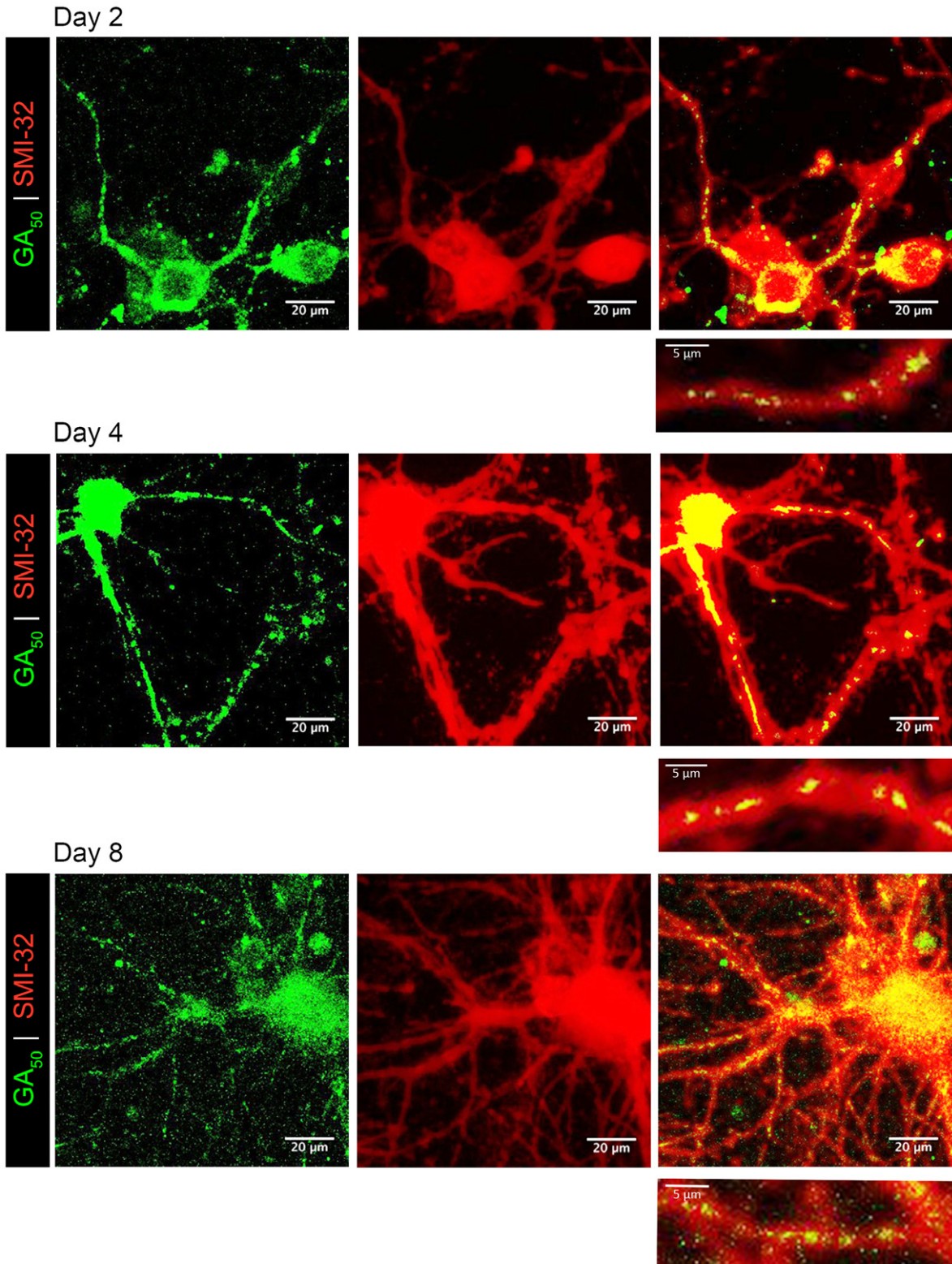

**Figure 1.  GA$_{50}$ aggregates are detected in neurites of cortical neurons over time.**

Primary rat cortical neurons transfected with eGFP-GA$_{50}$ were examined to determine at which time points preceding cell-death aggregates are found in neurites. Two days post-transfection, aggregates formed by eGFP-GA$_{50}$ (green) are detectable in neurites (SMI-32 staining in red). These aggregates remain localized to neurites at 96 h (4 days) and 288 h (8 days). Colocalization is indicated by yellow overlay of colors (right panels). Inset below each image shows enlargement of neurite regions containing aggregates, representative fields from 60× magnification z-stack confocal images, scale bar indicates 20 µm. Inset scale bars indicate 5 µm.

control cells at 8 days of eGFP expression (Fig EV1A). Quantification of the percentage of GA aggregate containing cells displaying neuritic aggregates revealed that at 2 days of expression 66.93% ± 5.57% of cortical GA-expressing neurons have aggregates in their neurites. Comparably, an identical transfection of mature motor neurons revealed that 55.62% ± 6.70% of GA-expressing motor neurons contain neuritic aggregates at 2 days of expression. Next, we wanted to determine the stability of GA aggregates over time. Mature cortical neurons (DIV10) were co-transfected with eGFP, or eGFP-GA$_{50}$ and a synapsin promoter-driven Td-tomato cell-filling reporter. After 24 h, transfected neurons were identified by co-positivity of eGFP and td-Tomato. The same neurons were then visualized every 24 h for 8 days. We observed that GA inclusions could be cleared over time by the neurons in which they have formed (Fig EV1B).

## GA aggregates are mobile within neurites

The presence of dynamic GA aggregates within neurons suggested additional possible deleterious consequences specific to neuronal cell lineages, including altered cellular trafficking of proteins and organelles to synaptic terminals, as well as dysfunctional synaptic transmission. To probe into these questions, we employed a variety of live-cell imaging functional assays and confocal imaging assessment of synaptic-associated proteins involved in neurotransmission.

Using high-resolution live-cell imaging to monitor trafficking of different cargoes within neurites, we first assessed whether the GA aggregates themselves were mobile or stationary, and whether the length of the GA repeat would be a determinant for such mobility. GA at different lengths was transfected into mature cortical or motor neurons at DIV7 and DIV5, respectively. The length range of eGFP-GA$_n$ utilized for our current experiments spanned from 25 to 400 repeats as previously described (Wen et al, 2014). The sequences for these constructs were designed following a randomized codon strategy to generate the specified poly-dipeptide sequence of the designated repeat length, but to avoid GGGGCC repeat expansions in the corresponding RNA transcripts (Wen et al, 2014). Brightfield image overlay with the eGFP channel clearly shows distinct GA aggregates within neuritic regions of both cell populations (Fig 2A and B). Following 48 h of expression, high-resolution 60× images were taken at a rapid frame rate, allowing the position of individual eGFP$^+$ GA particles to be tracked over time (NIH ImageJ software). In cortical neurons, aggregates of short GA repeat lengths (25–50) displayed higher mobility along neurites, while aggregates from longer GA repeats (100–400) were more stationary (**$P < 0.01$; Fig 2C). Similarly, a repeat length-dependent effect on GA particle velocity was observed in primary motor neurons (****$P < 0.0001$; Fig 2D). We next evaluated whether the presence of neuritic GA aggregates constituted an impediment to the normal trafficking of organelle cargoes along microtubules. To do so, we transfected cultures with eGFP-GA$_n$ repeats at increasing lengths and subsequently labeled mitochondria with the cell-permeable dye MitoTracker Deep Red, lysosomes with the cell-permeable dye Lysotracker Deep Red, or total RNA with SYTO RNASelect green. eGFP$^+$ neurons were identified, and mitochondria visualized using the 637-laser channel suitable for the excitation spectrum of the MitoTracker dye.

Representative images from cortical neurons demonstrate neuritic eGFP-GA aggregates and distinct mitochondrial labeling with the MitoTracker dye (Fig EV2A). In a similar fashion, eGFP$^+$ neurons were identified, and lysosomes visualized using the 637-laser channel suitable for the excitation spectrum of the Lysotracker dye. Representative images from cortical neurons demonstrate neuritic eGFP-GA aggregates and distinct lysosomal labeling (Fig EV2B). In contrast to the GA aggregates themselves, mitochondrial mobility was unaffected by the presence of GA inclusions within neurites. Mitochondrial mobility was compared across the GA length curve as well as with a GFP-only expressing control, showing no changes at 48 h in cortical neurons (Fig EV2C) or motor neurons (Fig EV2D). Performing the same analysis of lysosomal mobility revealed that only GA$_{400}$ containing cortical neurons displayed a significant reduction in mobility (*$P < 0.01$; Fig EV2E). We also assessed the mobility of total cellular RNA through the use of SYTO RNASelect dye labeling in mCherry control or mCherry-GA$_{50}$ expressing cortical neurons (Fig EV2F). Upon analysis, we again discovered no deficits in overall RNA mobility in GA-expressing cells. We have therefore established that GA aggregates move along cortical and motor neuron processes, without significantly impacting the kinetics of transport of cargoes vital for cell survival.

## GA-containing neurons display slow toxicity

We wanted to verify that these in vitro reported phenotypes were the result of GA-dependent mechanisms intrinsic to neurons, rather than a secondary phenomenon that is occurring in cells that are already committed to die following GA expression.

As such, we investigated the longitudinal survival of GA-expressing neurons. Mature cortical neurons (DIV7) or motor neurons (DIV5) were co-transfected with eGFP, eGFP-GA$_{50}$ or eGFP-GA$_{100}$, and a synapsin promoter-driven Td-tomato cell-filling reporter. After 24 h, transfected neurons were identified by co-positivity of eGFP and td-Tomato. The same neurons were then visualized every 24 h for an additional 13 days (cortical neurons) and 10 days (motor neurons). Expression of either GA$_{50}$ or GA$_{100}$ resulted in statistically significant decreased percent survival only at 13 days post-transfection (DIV23) (Fig 3A and B) and motor neuron survival at 10 days post-transfection (DIV15) (Fig 3C) compared with GFP-only expressing neurons. This translates in progressive loss of neurons over time, which can be clearly seen in representative images (marked with white arrows), and was confirmed by Kaplan–Meier survival curve analysis and log-rank testing of hundreds of neurons compiled over multiple independent experiments (****$P < 0.0001$; Fig 3). Hazard ratios at day 13 in cortical neurons revealed an increased hazard of death in both GA$_{50}$ (1.677) and GA$_{100}$ (1.263) conditions. Similarly, hazard ratios at day 10 in motor neurons revealed an increased hazard of death with both GA lengths of 1.345 and 1.736, respectively. While we observed a length-associated increase in toxicity in motor neurons, this was not the case in cortical neurons. We infer that this is due to the high burden and mobility of GA$_{50}$ aggregates in these cells. Therefore, the GA-induced cell death we observe occurs much later than the time frame in which we demonstrate mobile neuritic aggregates (2 days post-transfection). This suggests a window of time in which these aggregates have the potential to influence normal neuronal function.

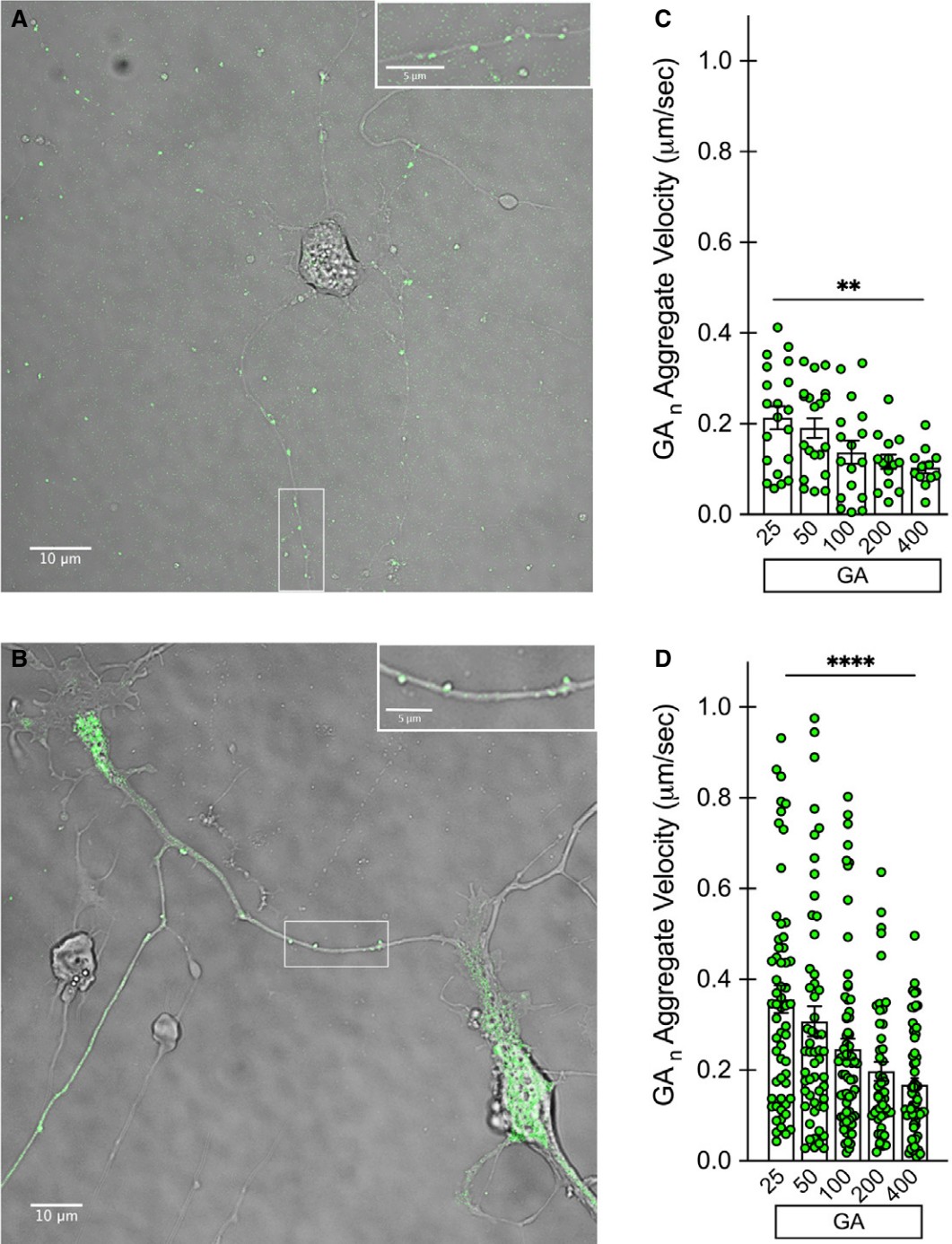

**Figure 2. Aggregates from short GA$_{25-50}$ repeat lengths are mobile, while those from longer repeats are more stationary.**

Primary rat cortical and motor neurons underwent live-cell imaging after 48 h of eGFP-GA$_n$ expression to determine aggregate mobility within neurites.

A   Example of a cortical neuron-expressing eGFP-GA$_{100}$ (green) with brightfield overlay, 60× magnification, scale bar indicates 10 μm. Inset in upper right shows enlargement of boxed area, to better visualize aggregates within neuronal processes. Inset scale bar indicates 5 μm.

B   Example of a motor neuron expressing eGFP-GA$_{25}$ (green) with brightfield overlay, 60× magnification, scale bar indicates 10 μm. Inset in upper right shows enlargement of boxed area, to better visualize aggregates within neuronal processes. Inset scale bar indicates 5 μm.

C   Quantification of eGFP-GAn aggregate velocity within cortical neuron neurites.

D   Quantification of eGFP-GAn aggregate velocity within motor neuron neurites.

Data information: Data presented as mean ± SEM. One-way ANOVA, *post hoc* Dunnett's multiple comparison test, **$P < 0.01$, ****$P < 0.0001$. Exact *P*-values can be found in Appendix Table S1. Velocity measurements from 10 aggregates were assessed from each of three biological replicates for each GA repeat length in each neuronal population.

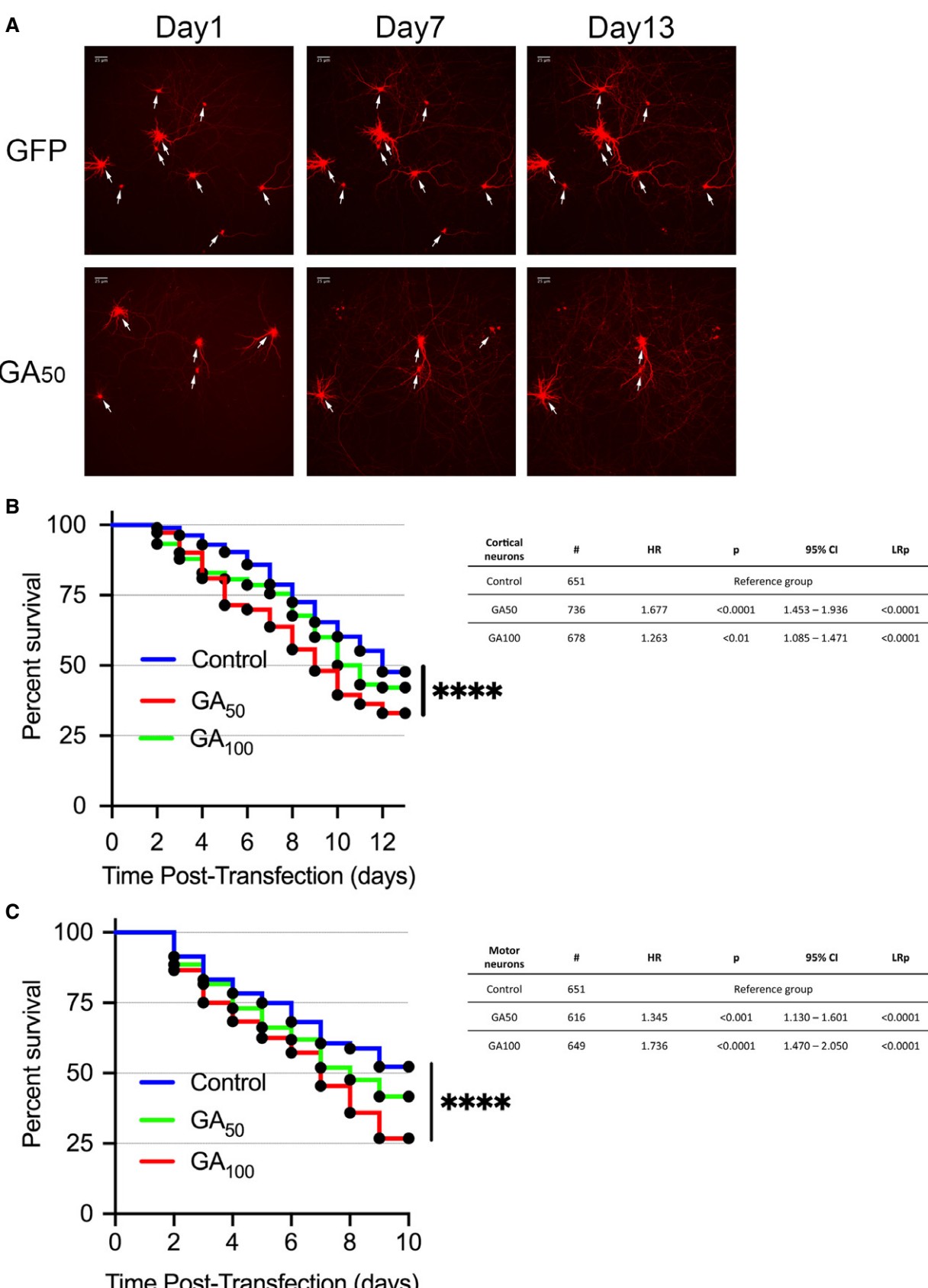

**Figure 3.**

**Figure 3. Poly-GA-containing neurons display delayed cell-death profile.**

Primary rat neurons were co-transfected with Td-tomato and eGFP-GA$_n$ plasmids or eGFP alone. The same neurons were imaged at 24-h intervals to assess viability.

A   Representative fields (right) of eGFP and eGFP-GA$_{50}$ cortical neurons at days 1, 7, and 13 post-transfection follow the survival of individual cells over time (white arrows). 20× magnification, scale bar indicates 25 μm.

B   Kaplan–Meier survival analysis of cortical neurons followed for 13 days post-transfection.

C   Kaplan–Meier survival analysis of motor neurons followed for 5 days post-transfection.

Data information: Statistical analysis performed by Cox-proportional hazards ration and Mantel–Cox log-rank tests. Tables showing neuron number (#), hazard ratio (HR), Cox-proportional hazards *P*-value (*P*), confidence interval (95% CI), and log-rank *P*-value (LRp) are shown to the right of each Kaplan Meier curve. At least 600 cells per condition were evaluated from a pooling of *n* = 6 independent experiments, ****P* < 0.0001. Exact *P*-values can be found in Appendix–Table S1.

## Synaptic release is compromised in neurons containing GA aggregates

A plausible outcome of GA inclusions within dendrites and axons is disturbance in synaptic neurotransmission. To assess the functional consequences of GA aggregates within neuronal processes, we resorted to live-cell imaging and stimulation paradigms to assess synaptic release and the length curve of eGFP-GA$_n$ constructs coupled with the cell-permeable dye FM4-64. After 48 h of eGFP-GA$_n$ expression, cells were loaded with this dye, which accumulates in synaptic vesicles and can be used as a readout for vesicle exocytosis (Gaffield & Betz, 2006; Verstreken *et al*, 2008) (Fig 4A). eGFP$^+$ cells were identified, and 10 synaptic puncta loaded with FM4-64 per region of interest per cell were monitored. Baseline measurements were acquired, followed by perfusion with high K$^+$ ACSF solution to induce synaptic firing and dye unloading (Fig 4B). Kinetics are presented as change (loss) in fluorescence (ΔF) over basal fluorescence (F), ΔF/F. A representative graph shows eGFP-GA$_{50}$ compared with eGFP expressing neurons over the course of stimulation (Fig 4C). Evaluation of cortical neurons expressing GA peptides revealed a striking abrogation of synaptic vesicle release. Compared to eGFP alone, which reduced fluorescence signal to 74.7% ± 3.3% of pre-stimulation levels, all of the GA lengths significantly impaired FM4-64 dye release (GA$_{25}$: 94.5% ± 1.6%, GA$_{50}$: 103.1% ± 1.7%, GA$_{100}$: 97.4% ± 2.8%, GA$_{200}$: 94.1% ± 2.5%, GA$_{400}$: 99.6% ± 0.5%) (***P* < 0.001). Furthermore, as analyzed by non-linear regression based on the plateau followed by one-phase decay model, there was a significant reduction in decay rate for cells containing GA aggregates. While the decay constant (k) for eGFP cells during stimulation was 0.497 s$^{-1}$, this was dramatically increased to 219.5 s$^{-1}$ in GA$_{50}$ containing cells. These results indicate that signals for neuronal firing are not being effectively executed in GA-containing cells; however, we could not yet rule out whether this was based on ineffective signaling or faults in synaptic release.

## Neurons with GA aggregates display increased Ca$^{2+}$ influx

It is well established that when sufficient depolarization occurs to trigger an action potential in a neuron, Ca$^{2+}$ influx is required for appropriate fusion of synaptic vesicles and release of neurotransmitter (Schneggenburger & Rosenmund, 2015). To determine whether the failure of synaptic unloading we observed in GA-containing cells was a result of altered Ca$^{2+}$ entry, a live-cell imaging paradigm was employed whereby mCherry-tagged GA constructs were co-transfected with a GCaMP Ca$^{2+}$ indicator. After 48 h of expression, co-transfected cells were selected for imaging based on dual mCherry and GCaMP signal (Fig 5A). Following basal GCaMP

fluorescence recording, neurons were perfused with high K$^+$ ACSF solution containing Ca$^{2+}$ (1.33 mM) to trigger neuronal depolarization. A robust surge in intracellular Ca$^{2+}$ signal was recorded following such stimulation (Fig 5A). Quantification of the maximum change in GCaMP fluorescence intensity over baseline was determined for each stimulated cell and compiled for 20 cells per condition. In both cortical and motor neurons, a significant increase in intracellular Ca$^{2+}$ signal was observed in mCherry-GA$_{50}$ containing cells compared with control mCherry-only expressing cells (*P* < 0.05, ***P* < 0.01; Fig 5B and C). We next sought to determine whether there was a GA length dependence to this increased Ca$^{2+}$ signal. Each of the GA length curve set of constructs was co-transfected with our GCaMP reporter construct. Following quantification of the maximum change in GCaMP fluorescence intensity over baseline for this series, we determined that while the GA lengths each produced significant increases in Ca$^{2+}$ influx, they did not significantly differ among themselves in the magnitude of this change (Fig EV3A). To determine whether the increased fluorescence signal was due to Ca$^{2+}$ influx into the neurons, stimulation of cells was performed in high K$^+$ ACSF that lacked free Ca$^{2+}$. In these conditions, no significant difference between control mCherry and mCherry-GA$_{50}$ containing cells was found, indicating that extracellular Ca$^{2+}$ entering into the cell was indeed responsible for the Ca$^{2+}$ surge inside the neurons (Fig 5B and C). Thus, in contrast to our expectations based on depressed synaptic unloading, we have found a significant increase in Ca$^{2+}$ influx in both cortical and motor neurons expressing GA dipeptides.

## Synaptic vesicle-associated protein 2 (SV2) is reduced in GA-expressing neurons *in vitro*

The aberrant increased Ca$^{2+}$ influx in GA-expressing neurons prompted us to examine whether synaptic proteins were being affected, as altered Ca$^{2+}$ homeostasis and signaling may have dramatic implications for synaptic release and vesicle fusion events (Zundorf & Reiser, 2011; Schneggenburger & Rosenmund, 2015). We investigated whether the levels of synaptic vesicle component proteins as well as neuritic morphology in our cortical and motor neuron *in vitro* paradigms were affected by the expression of eGFP-GA$_n$. The three proteins chosen for our initial exploration, synaptophysin, synaptic vesicle-associated protein 2 (SV2), and PSD-95, all serve vital purposes in synaptic transmission. SV2 is a crucial component of active zones and synaptic release machinery, as it forms complexes with other vesicle components synaptophysin, synaptotagmin, and synaptobrevin (Mutch *et al*, 2011; Kwon & Chapman, 2012; Li & Kavalali, 2017). While 3 isoforms of SV2 are encoded in the human genome, SV2a is the most widely distributed

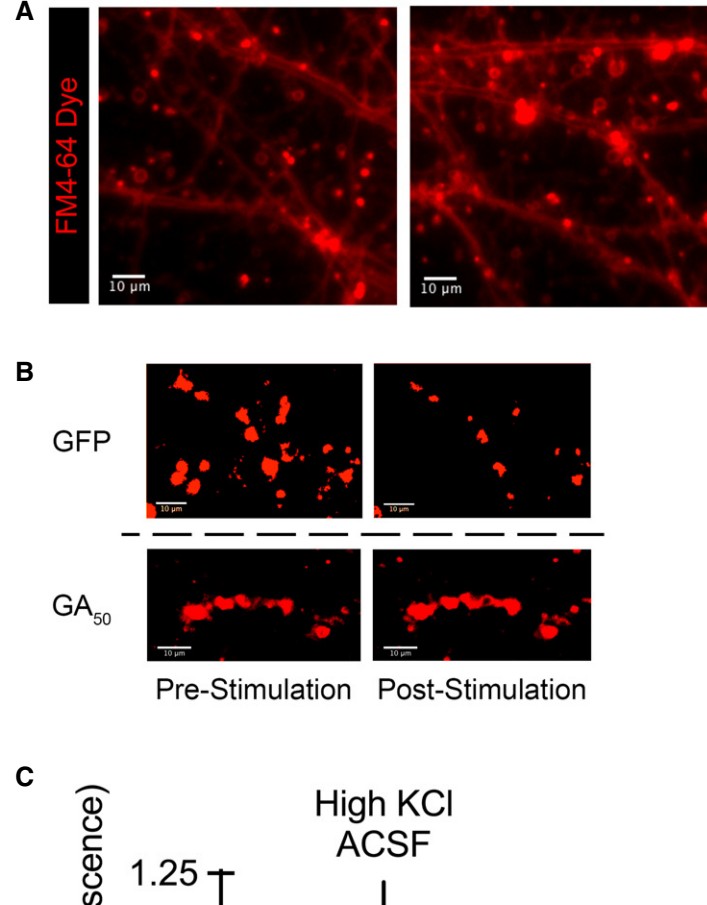

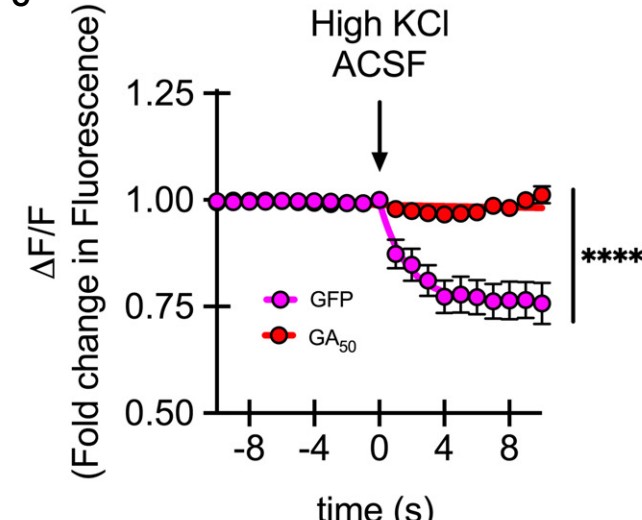

**Figure 4. GA aggregates significantly impair synaptic unloading.**

Cortical neurons were loaded with FM4-64 dye 48 h after transfection. Basal fluorescence was monitored prior to induced depolarization via perfusion with ACSF containing 50 mM KCl.

A   Representative dye loading showing defined puncta regions along neurites 40× magnification, scale bar indicates 10 μm.

B   Representative intensities of FM4-64 puncta regions pre- and post-stimulation demonstrate loss of synaptic puncta fluorescence intensity following stimulation in eGFP but not GA$_{50}$ containing neurons. These images visually reflect the process performed in an automated manner through our imaging software to determine puncta fluorescence intensity. An unbiased constant threshold is placed on each image per experiment; then fluorescence intensity is measured over time. The threshold applied isolates puncta from background, giving distinct regions. 8-bit images were thresholded and pseudo-color (red) was applied to enlarged puncta regions of interest, scale bar indicates 10 μm.

C   Graphical representation is the change in fluorescence (ΔF) normalized to basal fluorescence (F), ΔF/F. The final 10 s of baseline recording and 10 s of the depolarization phase are shown; arrow indicates start of depolarization. Non-linear regression was performed in GraphPad Prism7 using the plateau followed by one-phase decay model. The decay constant (k) for eGFP was 0.4971 s$^{-1}$, while for GA$_{50,}$ it was 219.5 s$^{-1}$. Similarly, the tau (τ) values were 2.012 s for eGFP and 0.00455 s for GA$_{50}$, respectively. Data presented as mean ± SEM for each imaged timepoint. Statistical significance was determined using the Sidak–Bonferroni method, ****$P < 0.0001$, $n = 20$ puncta regions from four biological replicates. Exact $P$-values can be found in Appendix Table S1.

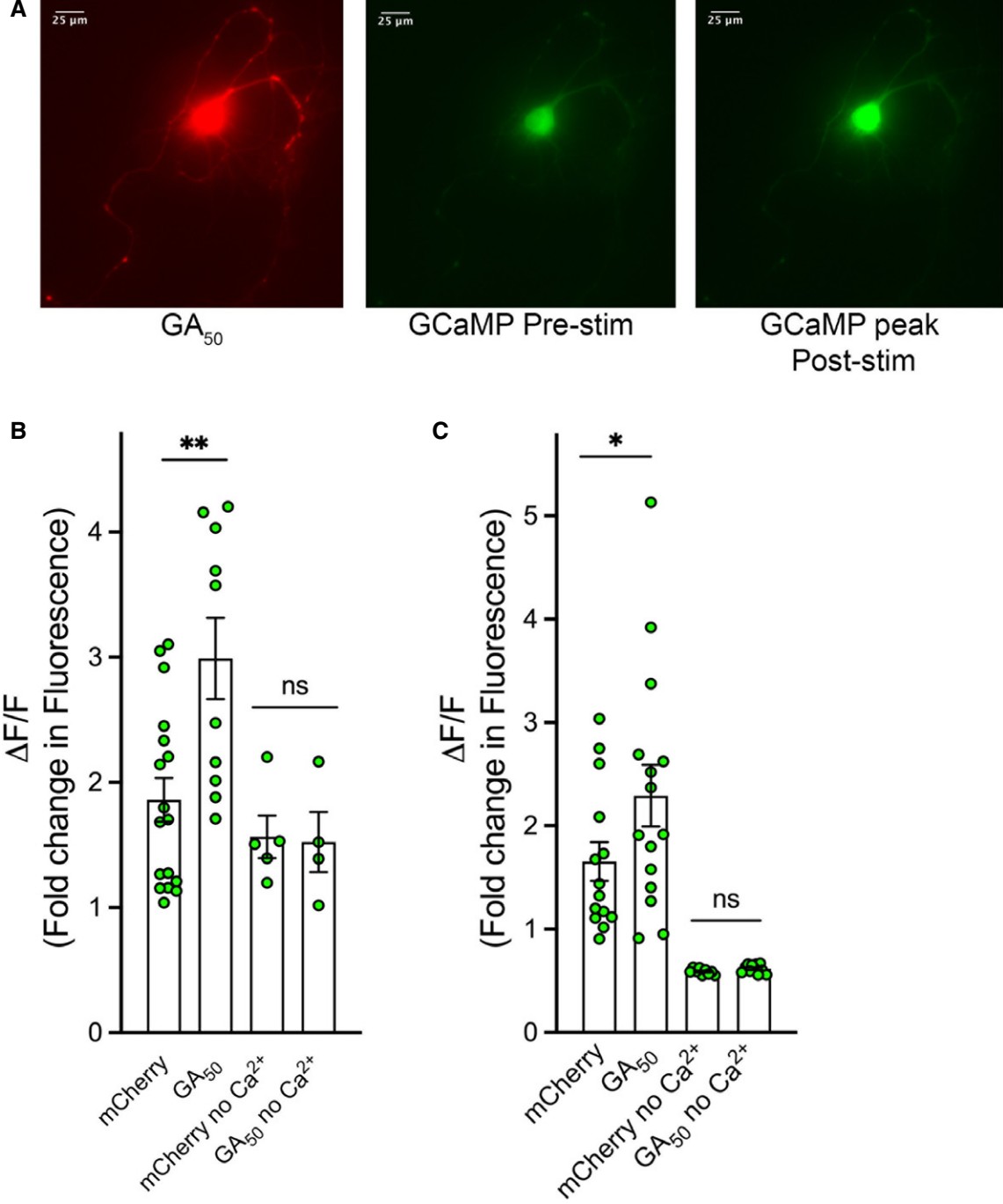

**Figure 5. Ca²⁺ influx is increased in neurons containing GA aggregates.**

Primary cortical or motor neurons were co-transfected with mCherry or mCherry-GA₅₀ and GCaMP6f. After 48 h, mCherry-positive cells were determined. Green fluorescence intensity was then recorded from identified neurons. Basal fluorescence was monitored prior to induced depolarization via perfusion with ACSF containing 50 mM KCl.

A Representative images of a co-transfected cortical neuron expressing mCherry-GA₅₀ (left panel) and GCaMP6f (right two panels). This image shows a representative demonstration of the increase in detectable green fluorescence from the GCaMP6f indicator pre- and post-stimulation (center and right, respectively). 40× magnification, scale bar indicates 25 μm.

B, C Graphical representation of the peak change in fluorescence (ΔF) following ACSF perfusion normalized to basal fluorescence (F), ΔF/F. The contribution of Ca²⁺ influx was determined by experiments in which stimulation was performed in ACSF containing high KCl but no Ca²⁺. Quantification of the peak change in fluorescence in cortical neurons (B) and motor neurons (C).

Data information: Data presented as mean ± SEM. One-way ANOVA with uncorrected Fisher's LSD test, *$P < 0.05$, **$P < 0.01$. A total of 20 cells per condition were pooled from eight independent experiments. Exact *P*-values can be found in Appendix Table S1.

in the brain and is the target of several pharmaceutical compounds (Crevecoeur *et al*, 2013). This isoform has been the focus of our experiments, and will be referred to as simply SV2 throughout this article. SV2 is also speculated to be a transport protein for neurotransmitter import into synaptic vesicles (Feany *et al*, 1992) and plays an integral role in maintaining appropriate machinery for $Ca^{2+}$ entry at synapses through interactions with voltage-gated $Ca^{2+}$ channels (Vogl *et al*, 2015). Additionally, through association with laminin and laminin-related proteins, SV2 specifically participates in synapse re-innervation (Son *et al*, 2000; Singhal & Martin, 2011). Synaptophysin is also a crucial component of synaptic vesicles and one of the most abundant proteins, making up 8% of total vesicle content (Evans & Cousin, 2005). Instead of facilitating evoked neurotransmitter release, synaptophysin is believed to play a role in vesicular recycling (Evans & Cousin, 2005). An equally important region to evaluate is the postsynaptic density, as this is where signal transduction events are relayed. PSD-95 is a key structural component of the postsynaptic density and serves as our marker of potential changes in this cellular region. It is thought that PSD-95 plays a role in the organization of the postsynaptic density, through binding of constituent proteins like AMPA receptors, NMDA receptors, adhesion molecules, and other scaffolding proteins (Chen *et al*, 2011). Mature cortical and motor neurons were transfected with our eGFP-$GA_n$ construct at different lengths and immunostained for these synaptic proteins. Visualization by confocal microscopy followed by counting of positive puncta per eGFP$^+$ cell revealed a significant GA repeat length-dependent reduction in SV2 levels in both cortical and motor neurons (****$P < 0.0001$; Fig 6A–D). Total neurite length was unaffected by expression of GA at any repeat length, ensuring that this was a reduction of synaptic protein content per neurite length rather than a reduction in overall neurite area (Fig 6C and D; right panels). Synaptophysin and PSD-95 levels were unchanged when compared with eGFP alone or across the GA length curve (Fig EV3B and C). Overall, among the synaptic proteins examined, a specific, and significant length-dependent reduction of SV2 was observed both in primary cortical and motor neurons. We have begun to address potential mechanistic links between GA expression and SV2 altered levels and localization, by assessing GA colocalization with SV2 protein and mRNA. In cortical neurons transfected with our mCherry-$GA_{50}$ construct and stained as mentioned above at 2 days of expression, we also evaluated GA colocalization with SV2 protein within SMI-32 neuritic regions (Fig EV4A). Quantification across the GA length curve revealed that with all GA lengths, a small percentage of SV2 protein has been sequestered into GA aggregates, approximately 3.5% of total cellular SV2. In order to assess possible interactions of GA with SV2 mRNA, a custom beacon probe was generated against SV2 mRNA (Bio-Synthesis, Inc.). This beacon was co-transfected along with our eGFP-$GA_{50}$ construct, and cells were evaluated for colocalization following 2 days of GA expression (Fig EV4B). We found that $20.97\% \pm 2.67\%$ of the RNA beacon in GA-expressing cells was colocalized with GA. These intriguing findings will help guide extensive in-depth evaluations of GA interactions in future studies.

## SV2 is reduced in C9orf72-ALS patient-derived iPS neurons

We next wished to validate that our findings will be physiologically relevant to C9orf72-ALS patients. We obtained i3 iPS lines of a

C9orf72-ALS patient, and the paired isogenic line which had the GGGGCC repeat expansion removed (Fernandopulle *et al*, 2018). These cells were expanded and differentiated into cortical neurons according to their previously defined protocol and allowed to mature for 21 days *in vitro*. At this time, cells were analyzed *via* immunofluorescence, qPCR, and immunoblotting. Cortical i3 neurons on coverslips were probed for either SV2 or synaptophysin and counterstained with neurofilament. Visualization by confocal microscopy was followed by quantification of the synaptic protein colocalized with neurofilament. This analysis revealed that SV2 was again specifically reduced in neurons harboring a C9orf72 repeat expansion compared with an isogenic control (*$P < 0.05$; Fig 7A and C), while synaptophysin levels remained unaltered (Fig 7B and D). We also tested the overall mRNA levels of SV2 at this time point in the two i3 lines. In the C9 line, SV2 mRNA was reduced to 0.249-fold compared with the paired isogenic line (**$P < 0.01$; Fig 7E). We also assessed SV2 at the protein level by preparing cellular lysates and immunoblotting these samples with a probe for SV2. Following densitometric analysis compared with total protein loading, we found that SV2 protein was reduced to 0.578-fold compared with the isogenic line (*$P < 0.05$; Fig 7F). To assess whether these findings would also be applicable to iPS-derived motor neurons, cell pellets were procured from 32-day differentiated control and C9orf72-ALS iMNs via the Answer ALS Consortium. Immunoblotting of samples derived from these whole-cell lysates with probes for SV2 and synaptophysin revealed a trend toward reduction in SV2 levels ($P = 0.1116$) (Fig 7G and H). Notably, this analysis was done with genetically unrelated C9orf72-ALS and control lines, using a protocol that produces a mixed neuronal population with 20–30% Islet1$^+$ motor neurons. It is likely that if the efficiency of differentiation could be enhanced the reduction of SV2 would be even more pronounced. Overall, these findings from patient-derived iPSC lines have supported our SV2 finding in the context of the total cellular consequences resulting from the C9orf72 repeat expansion.

## Replenish SV2 restores synaptic function and attenuates cellular toxicity

We next sought to establish whether the reduction in SV2 levels in GA-expressing cells was a viable molecular candidate for therapeutic intervention, through restoration of SV2 levels via transduction of cortical and motor neurons with lentivirus. We obtained the rSV2a-eGFP-pRRL plasmid construct, which has previously been shown to robustly express rat SV2a *in vitro* (Yao *et al*, 2010). Following generation of SV2 lentivirus from this construct, we infected mature cortical neurons and evaluated them following 4 days of expression for SV2 upregulation at the mRNA and protein levels by qPCR and immunoblotting, respectively. Compared with non-transduced control cells, we observed a significant 1.99-fold increase of SV2 mRNA (**$P < 0.01$), as well as a significant 1.72-fold increase of SV2 protein (***$P < 0.001$; Fig EV4C and D).

We chose to evaluate any rescue phenotypes using $GA_{50}$ as our representative GA length, since it displayed robust cellular effects *in vitro*. Initially, we examined whether SV2 puncta levels were restored in $GA_{50}$ containing neurons transduced with SV2$_{pRRL}$ through immunofluorescent staining and analysis following 2 days of expression. We found that the significant reduction of SV2 puncta observed in GA-containing cells was normalized to control levels

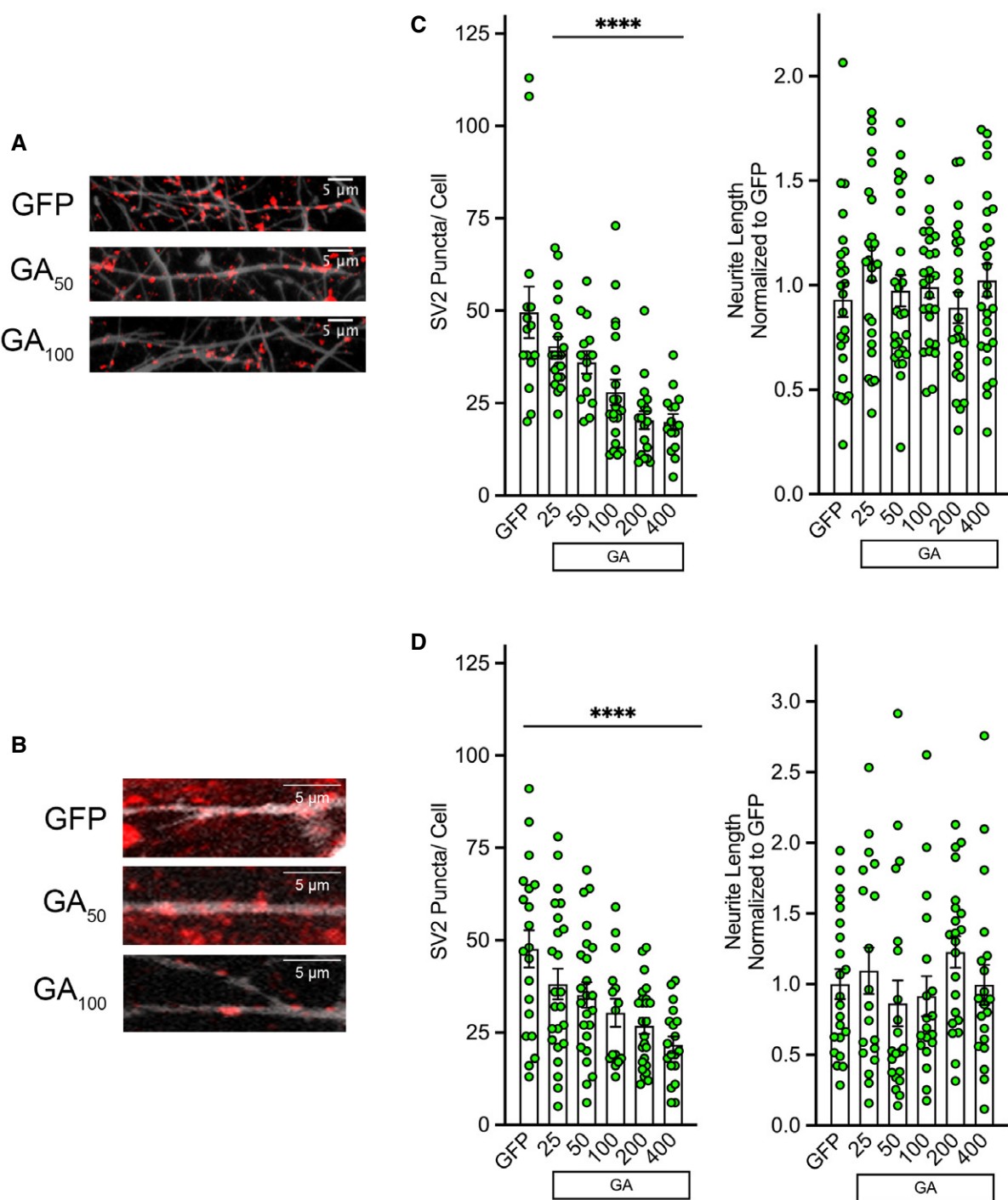

**Figure 6. Synaptic vesicle component SV2 is reduced in GA aggregate containing neurons.**

eGFP-GA$_n$ dipeptides were expressed in primary cortical or motor neurons for 48 h, and then immunostained for neurofilament (gray) and SV2 (red).

A, B   Representative z-stack confocal images of neurites from eGFP, eGFP-GA$_{50}$, and eGFP-GA100 expressing cortical (A) or motor (B) neurons. 60× magnification, scale bar indicates 5 μm.

C, D   Quantification of SV2 puncta along neurites in cortical neurons (C) or motor neurons (D) by ImageJ manual counting, normalized to eGFP-only expressing cells. (C) Cortical neurons demonstrated a significant GA length-dependent reduction, as well as a reduction in all cells containing GA repeats greater than 50 (left panel). (D) Motor neurons demonstrated a significant GA length-dependent reduction, as well as a reduction in all cells containing GA repeats greater than 100 (left panel). Evaluation of neurite length by NeuronJ software revealed no differences in total neurite length per cell in cortical neurons (C) or motor neurons (D) expressing any GA repeat length compared with eGFP alone (right panels).

Data information: Data presented as mean ± SEM. One-way ANOVA, *post hoc* Dunnett's multiple comparison test, ****$P < 0.0001$, $n = 3$ independent biological replicates, $m = 5–7$ cells per replicate. Exact $P$-values can be found in Appendix Table S1.

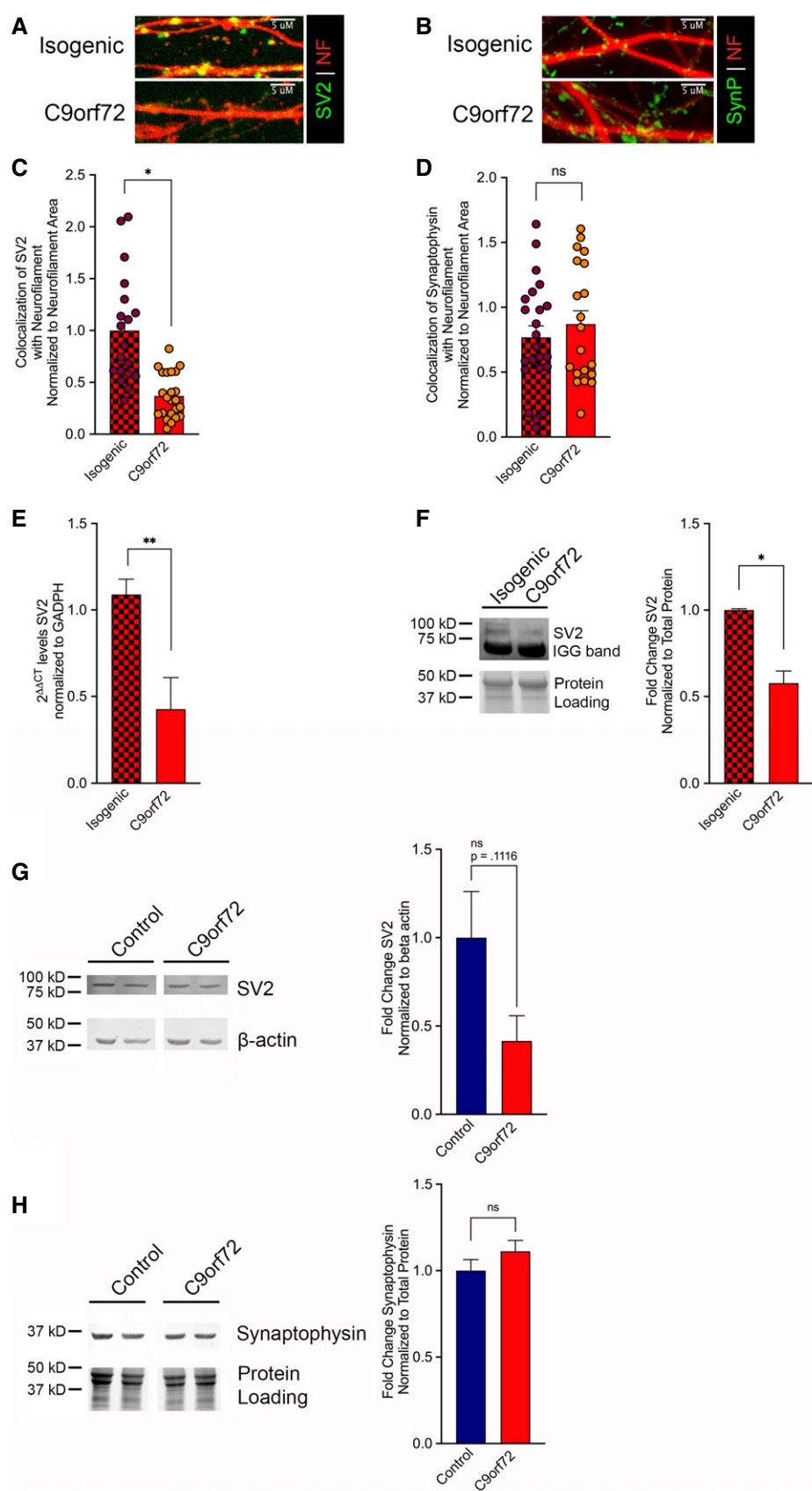

Figure 7.

**Figure 7.  SV2 is reduced in i3 neurons from a C9orf72-ALS patient.**

A, B     i3 cortical neurons from a C9orf72-ALS patient and the paired isogenic control were differentiated for 21 days then were immunostained for neurofilament (red), and SV2 (green) (A) or synaptophysin (green) (B). Shown are representative z-stack confocal images of neurites.

C, D     Quantification of SV2 (C) or synaptophysin (D) colocalized with neurofilament normalized to neurofilament area. This analysis revealed a significant reduction in SV2 levels in the C9orf72 line compared with the isogenic control, while synaptophysin remained unaltered. Data presented as mean ± SEM. Unpaired *t*-test, *$P < 0.05$. 15 non-overlapping regions from three differentiated wells for each genotype were evaluated.

E        mRNA was Trizol extracted from three independent wells of C9-ALS and isogenic i3 cortical neurons. RNA was converted to cDNA using the Superscript First-strand kit, and qPCR was performed using PowerUp SYBR Green. Measurements were normalized to the housekeeping gene GAPDH and then to isogenic levels. Analysis was performed using the ΔΔCT method. $2^{\Delta\Delta CT} \pm$ SE is presented. Unpaired *t*-test, **$P < 0.01$.

F        Whole-cell lysates were generated from three independent wells of C9-ALS and isogenic i3 cortical neurons differentiated for 21 days. Lysates were immunoblotted for SV2 and normalized to total protein loading. Quantification of band intensities revealed a significant decrease in SV2 protein levels in the C9orf72 lines compared with the isogenic. Data presented as mean ± SEM. Unpaired *t*-test, *$P < 0.05$.

G, H     Whole-cell lysates were generated from iMNs differentiated for 32 days. Lysates were immunoblotted for SV2 (G) or synaptophysin (H). Loading controls were β-actin or total protein loading. Quantification band intensities normalized to loading controls revealed a trend toward reduction in SV2 protein levels in the C9orf72 cell lines compared with controls, while synaptophysin levels remained unaltered. Data presented as mean ± SEM. Unpaired *t*-test. Samples from five independent control and four independent C9orf72 patient lines were used. Exact *P*-values can be found in Appendix Table S1.

Source data are available online for this figure.

through this exogenous SV2 expression (Fig EV4E and F). While SV2 levels in $GA_{50}$ containing cells were still significantly lower than control cells transduced with $SV2_{pRRL}$ (****$P < 0.0001$), levels that were comparable to normal control neurons suggested the potential for recovery of synaptic function. Using the same array of live-cell imaging techniques through which we initially detailed disruption of synaptic release, altered $Ca^{2+}$ influx, and neuronal toxicity, we next determined whether restoration of SV2 levels would be capable of rescuing these cellular defects. First, mature cortical neurons were transfected with eGFP or eGFP-$GA_{50}$ and co-transduced with $SV2_{pRRL}$ lentivirus. Following 2 days of expression, synaptic release was again measured using our FM4-64 dye paradigm. When neurons co-expressed $GA_{50} + SV2_{pRRL}$, the abrogation of synaptic release in cells expressing $GA_{50}$ alone was restored to comparable levels of that of control GFP neurons (***$P < 0.001$; Fig 8A). Calcium influx was also evaluated at 2 days following co-transfection of mCherry-$GA_{50}$ and gCAMP plus transduction with $SV2_{pRRL}$. Similar to the synaptic release results, when neurons co-expressed $GA_{50} + SV2_{pRRL}$, the elevation of $Ca^{2+}$ influx caused expression of

$GA_{50}$ alone was restored to comparable levels of that of control neurons (***$P < 0.001$; Fig 8B). Finally, cellular toxicity was longitudinally evaluated in both cortical and motor neurons to determine if restoration of SV2 levels would impact the cellular toxicity caused by GA. Following the same principle as our Fig 3 longitudinal survival assay, mature cortical and motor neurons were co-transfected with mCherry or mCherry-$GA_n$ and assessed for survival over time in the presence or absence of SV2 restoration via $SV2_{pRRL}$ lentiviral transduction. In cortical neurons, the significant cellular toxicity of $GA_{50}$ was rescued by restored levels of SV2 via expression of $SV2_{pRRL}$, with the hazard ratio dropping from 1.473 for $GA_{50}$ to 0.7277 for $SV2_{pRRL}$ co-expression (****$P < 0.0001$) (Fig 8C). As a control, the toxic C9orf72 dipeptide $PR_{50}$ was also expressed in neurons with or without $SV2_{pRRL}$ co-expression to determine if this was a specific GA-mediated result. Assessment of survival of $PR_{50}$ containing neurons revealed that significant reduction in survival was observed with or without $SV2_{pRRL}$, with hazard ratios of 1.772 for $PR_{50}$ and 1.784 for $PR_{50} + SV2_{pRRL}$, respectively (****$P < 0.0001$) (Fig 8D). Finally, we also evaluated motor neuron

**Figure 8.  GA-induced synaptic dysfunction and cellular toxicity can be rescued by replenishment of SV2.**

A        Cortical neurons were loaded with FM4-64 dye 48 h after transfection with eGFP-$GA_{50}$ and transduction with rSV2a-eGFP-pRRL lentivirus. Basal fluorescence was monitored prior to induced depolarization via perfusion with ACSF containing 50 mM KCl. Graphical representation is the change in fluorescence (ΔF) normalized to basal fluorescence (F), ΔF/F. The final 60 s of baseline recording and 60 s of the depolarization phase are shown; arrow indicates start of depolarization. The lack of synaptic release in the case of $GA_{50}$ expression (red line) was rescued back to levels comparable to eGFP controls (pink line) in the case of co-expression of rSV2a-eGFP-pRRL (orange line). Statistical significance was determined using the Sidak–Bonferroni method, ***$P < 0.001$, $n = 10$ puncta regions from each of 4 biological replicates.

B        Cortical neurons were co-transfected with mCherry or mCherry-$GA_{50}$ and GCaMP6f, then transduced with rSV2a-eGFP-pRRL. Green fluorescence intensity was recorded from identified mCherry-positive neurons. Basal fluorescence was monitored prior to induced depolarization via perfusion with ACSF containing 50 mM KCl. Quantification of the peak change in fluorescence (ΔF) following ACSF perfusion normalized to basal fluorescence (F), ΔF/F. The significant increase in $Ca^{2+}$ influx observed following $GA_{50}$ expression (*$P < 0.05$) is resolved when rSV2a-eGFP-pRRL is co-expressed. Data presented as mean ± SEM. One-way ANOVA with Sidak's multiple comparisons test, at least 10 cells per condition, pooled from $n = 3$ independent biological experiments. No significant differences were observed between mCherry or mCherry + SV2 conditions; groups were merged for this analysis.

C–E      Neurons were co-transfected with mCherry-$GA_n$ plasmids or mCherry alone and transduced with rSV2a-eGFP-pRRL and then imaged at 24-hour intervals. (C) Kaplan–Meier survival analysis of cortical neurons expressing $GA_{50}$ or $GA_{50} + SV2$ followed for 13 days. The significant cellular toxicity induced by $GA_{50}$ is rescued by SV2 replenishment. While the hazard ratio of $GA_{50}$ was 1.473, this was decreased to 0.7227 for $GA_{50} + SV2$. At least 190 cells per condition were evaluated from a pooling of $n = 3$ independent experiments, log-rank Mantel–Cox test, ****$P < 0.0001$. No significant differences were observed between eGFP or eGFP + SV2 conditions, and groups were merged for this analysis as control. (D) A Kaplan–Meier survival analysis was performed for $PR_{50}$ and $PR_{50} + SV2$ containing cells. Under these conditions, significant toxicity is observed regardless of increased SV2 levels, with the hazard ratio of $PR_{50}$ containing cells (1.772) remaining high for those expressing $PR_{50} + SV2$ (1.784). At least 150 cells per condition were evaluated from a pooling of $n = 3$ independent experiments, log-rank Mantel–Cox test, ****$P < 0.0001$. No significant differences were observed between eGFP or eGFP + SV2 conditions, and groups were merged for this analysis as control. (E) Kaplan–Meier survival analysis of motor neurons expressing $GA_{50}$ or $GA_{50} + SV2$ followed for 10 days. The significant cellular toxicity induced by $GA_{50}$ is rescued by SV2 replenishment. While the hazard ratio of $GA_{50}$ was 1.345, this was decreased to 0.7769 for $GA_{50} + SV2$. At least 200 cells per condition were evaluated from a pooling of $n = 3$ independent experiments, log-rank Mantel–Cox test, ****$P < 0.0001$. All exact *P*-values can be found in Appendix Table S1.

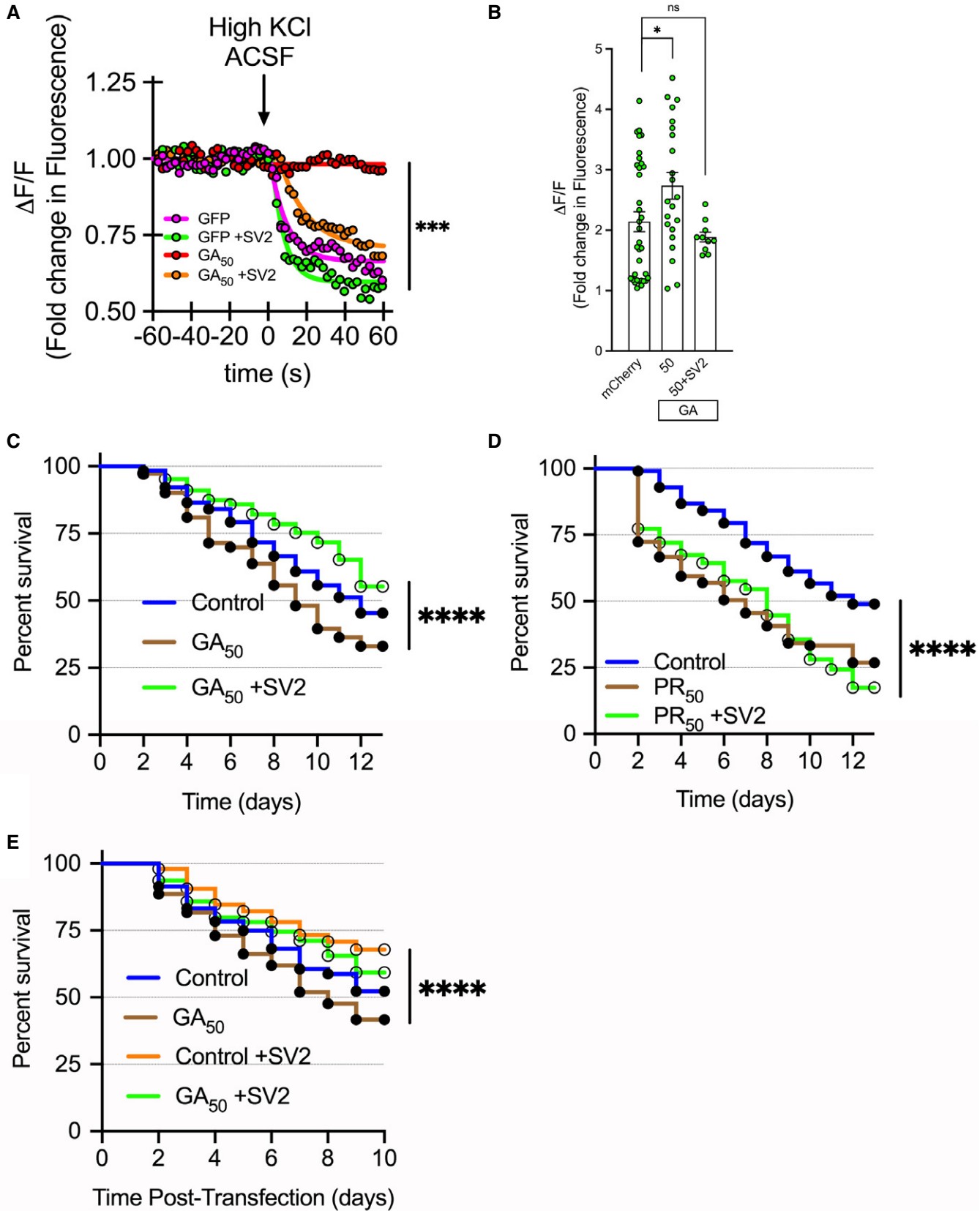

**Figure 8.**

longitudinal survival in cells expressing GA constructs as well as the SV2$_{pRRL}$ rescue construct. The significant toxicity of GA$_{50}$ (hazard ratio of 1.345) was once again rescued by SV2$_{pRRL}$ co-expression (hazard ratio 0.7769) (****$P < 0.0001$) (Fig 8E). Overall these findings suggest that despite a reduction of SV2 in the presence of GA dipeptides, intervening in such a way so as to restore normal SV2 levels is sufficient to recover synaptic transmission and synaptic signaling events as well as to prevent cell death.

### Transgenic GA$_{149}$-CFP mice display a specific reduction in synaptic component SV2

After investigating the repercussions of GA expression on neuronal function *in vitro*, we turned to our animal model to determine whether similar synaptic consequences also occur *in vivo*. As we have previously described, our transgenic mice express a 149-repeat GA dipeptide, coupled to a cyan fluorescent protein tag (GA$_{149}$-CFP) exclusively in neurons of the spinal cord, brainstem, and deep cerebellar nuclei (Schludi *et al*, 2017). Age-dependent accumulation of GA inclusions in these regions results in progressive impairments in gait and balance, without reduction in motor neuron number, suggesting impaired neuromuscular transmission.

To determine the effects of neuronal GA expression on synaptic components, levels of SV2, synaptophysin, and PSD-95 were assessed by immunoblotting samples derived from spinal cords of these animals. In whole-cell lysates generated from 20-month-old mice, analysis revealed a significant reduction of SV2 protein in transgenic animals compared with wild-type controls (*$P < 0.05$; Fig 9A), whereas synaptophysin and PSD-95 levels were not significantly altered (Fig 9B and C). Immunohistochemical staining from cervical sections of paraffin-embedded spinal cords from paired 20-month-old wild-type and transgenic animals revealed no differences in motor neuron number by ChAT staining or by neuronal DAPI morphology analysis (data presented as mean ± SEM. Unpaired *t*-test. $n = 4$ animals per group; Table 1). Additionally, immunohistochemical staining was performed on sections derived from snap-frozen hindlimbs within the cohort of 20-month-old mice. A dramatic reduction of SV2 levels in proximity to neuromuscular junctions was observed in transgenic GA$_{149}$ animals compared to wild type (Fig 10A). Assessment of synaptic proteins SV2 and synaptophysin at the NMJ confirmed a specific reduction of SV2 in transgenic animals compared with controls (*$P < 0.05$), while again synaptophysin levels were unaltered (Fig 10B and C). Thus, from tissue derived from GA$_{149}$ animals, we have confirmed, *in vivo*, a specific reduction of SV2 protein prior to cell death of affected spinal cord motor neurons.

## Discussion

Loss of functional communication between presynaptic motor neurons and postsynaptic muscle is the hallmark feature of all forms of ALS and has been advocated as a promising target for therapeutic intervention. In a recent report using a mouse model expressing the ALS-causative TDP-43$_{Q331K}$ mutation, neurotransmission deficits in the form of defects in synaptic vesicle fusion and release, decreased probability of release, and reduced quantal content of vesicles were described 7 months prior to motor neuron loss (Chand *et al*, 2018).

The disease causative GGGGCC hexanucleotide repeat expansion in C9orf72 is no exception to this principle; in fact, early synaptic deficits have now been identified in paradigms modeling this mutation and have been suggested as potential targetable disease features (Sareen *et al*, 2013; Devlin *et al*, 2015). Previous reports in two independent *Drosophila melanogaster* models have investigated structural features of the neuromuscular junction in flies expressing 30 or more GGGGCC repeats. These groups contributed valuable insight into C9orf72-ALS synaptic pathology, as both reported significant reductions in active zone number (Freibaum *et al*, 2015; Zhang *et al*, 2015). Importantly, they also demonstrated significant decreases in synaptic bouton number (Freibaum *et al*, 2015) as well as attenuation of evoked potentials due to reduced synaptic quantal content (Zhang *et al*, 2015). As vitally important as these studies have been for our understanding, the outcomes of expanded GGGGCC repeat stretches expressed in the fly could be resultant from either repeat expansion RNA, dipeptide repeat proteins (DPRs), or a synergistic combination of both. Within this study, we pinpointed our focus toward DPR-mediated effects.

In agreement with several other groups, our previous findings demonstrated that glycine-arginine (GR) and proline-arginine (PR) dipeptides were rapidly and robustly toxic to primary cortical and motor neurons (Mizielinska *et al*, 2014; Wen *et al*, 2014; Boeynaems *et al*, 2016). In this model, when cortical neurons were examined to 7 days or motor neurons to 3 days post-expression, GA up to 400 dipeptide repeats did not yet precipitate neuronal cell death (Wen *et al*, 2014). In our transgenic mouse model, which drives GA$_{149}$-CFP off of the Thy-1 promoter leading to expression in spinal cord and brainstem, we have observed pronounced motor deficits that correlate with increasing burden of GA inclusions, but with a lack of motor neuron dropout (Schludi *et al*, 2017). These results were intriguing as work examining GA effects in maturing neurons demonstrated reduced dendritic complexity and cell death (May *et al*, 2014). Additionally, a second mouse model was published which confines GA$_{50}$ expression to the cortex. While this model also develops motor and cognitive behavioral deficits by 6 months of age, concomitant neuronal loss and brain atrophy also occurred (Liu *et al*, 2016). These conflicting findings and the possibility that motor deficits and neuronal dysfunction may precede neuronal cell death led us to thoroughly investigate the nuances of GA expression in mature neurons *in vitro* and ultimately led to our examination of synaptic transmission and synaptic proteins.

We discovered that in addition to cytosolic inclusions, GA aggregates were also present within neuritic regions and were not stationary but rather displayed different degrees of mobility in a repeat length-dependent fashion. In response to a stimulus to evoke synaptic transmission, functional analysis revealed that synaptic unloading was significantly attenuated in GA-expressing neurons. By investigating mechanisms by which this may occur, we uncovered that both cortical and motor neurons containing GA aggregates demonstrated significant increases in Ca$^{2+}$ influx compared to control neurons upon depolarizing conditions. Probing the levels of synaptic proteins revealed that synaptic vesicle-associated protein 2, SV2, was specifically reduced. This specific reduction in SV2 was confirmed in i3 differentiated neurons carrying a C9orf72 GGGGCC repeat expansion, suggesting that this finding will be applicable to patients carrying the full complement of C9orf72 expansion-

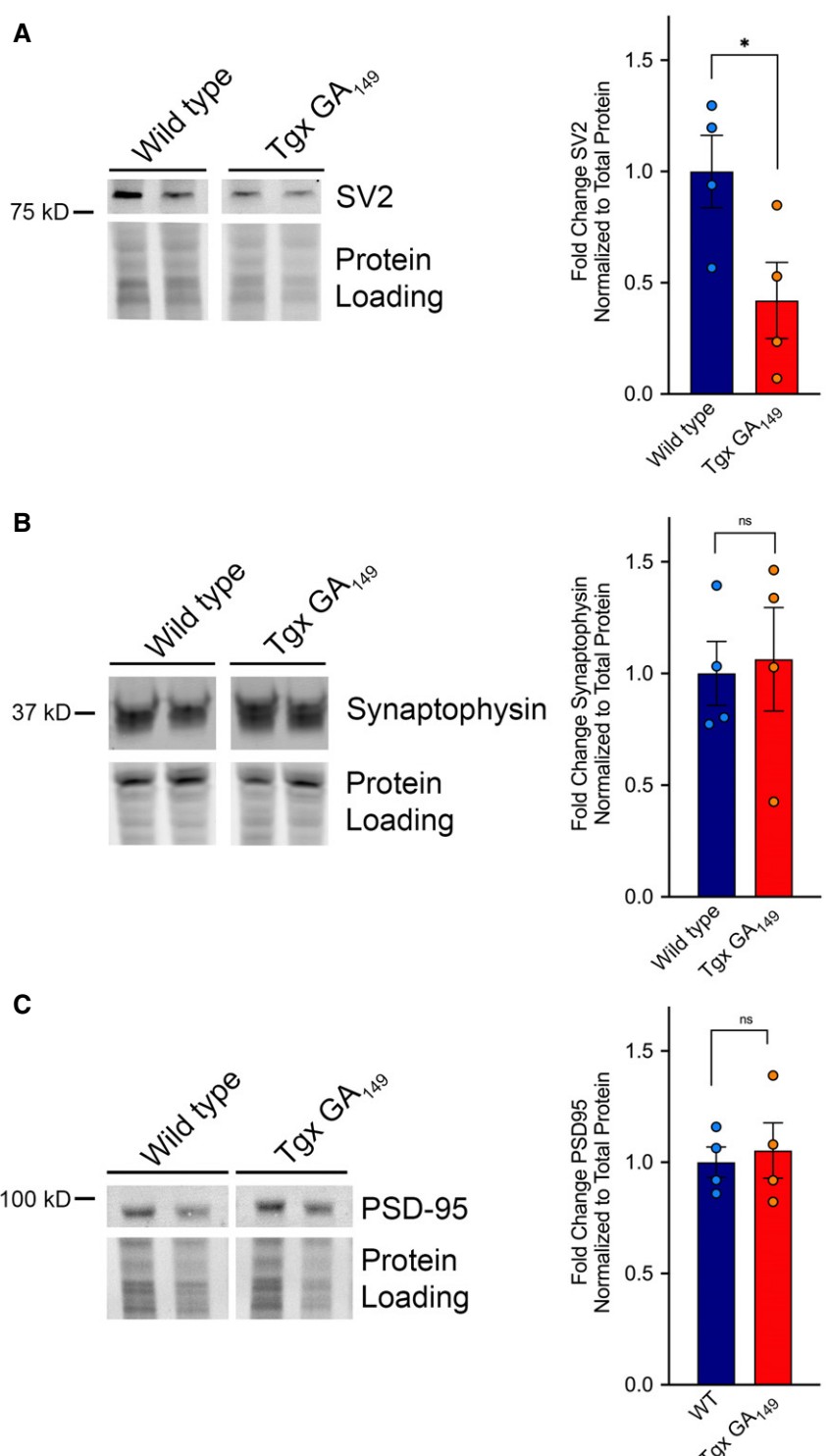

**Figure 9. SV2 is specifically reduced in spinal cord lysates of symptomatic transgenic GA149-CFP mice.**

Spinal cord lysates were generated from fresh-frozen tissue of 20-month-old mice either expressing $GA_{149}$-CFP under the Thy1 promoter (Tgx $GA_{149}$) or wild-type controls.

A–C  Lysates were immunoblotted for SV2 (A), synaptophysin (B), and PSD-95 (C). Total protein levels were determined by imaging the Mini-PROTEAN TGX stain-free gel prior to transfer. Quantification of band intensities normalized to total protein loading revealed a significant reduction in SV2 protein levels in the GA-expressing cohort compared with wild-type animals, while synaptophysin and PSD-95 levels remained unaltered. Data presented as mean $\pm$ SEM. Unpaired *t*-test, *$P$ < 0.05. A total of four animals per group were examined. Exact *P*-value can be found in Appendix Table S1.

Source data are available online for this figure.

Table 1.   Motor neuron counts by immunocytochemical staining.

|                          | ChAT[+]         | DAPI[+]          |
| ------------------------ | --------------- | ---------------- |
| Wild type                | $14.50 \pm 1.50$ | $16.50 \pm 2.53$ |
| $GA_{149}$-CFP           | $14.25 \pm 1.37$ | $15.38 \pm 1.00$ |

mediated cellular toxicities. While we found that cortical and motor neurons do eventually succumb to GA-induced cell death (at time points extended past our originally published assessment (Wen *et al*, 2014)), the synaptic changes we have described occur several days prior to cellular demise. Notably, when we employed a strategy for restoration of SV2 levels in GA-expressing cells, we were able to functionally restore synaptic release, prevent aberrant $Ca^{2+}$ signaling, and rescue GA-induced cellular toxicity. Finally, we returned to our *in vivo* model, wherein we validated our observation of a specific reduction in SV2 levels. Despite being aged to 20 months, we still did not observe a reduction in spinal cord motor neuron number in transgenic $GA_{149}$-CFP mice compared with age-matched controls.

In our models, SV2 levels are diminished *in vitro* and *in vivo*. SV2, also known as synaptic vesicle glycoprotein 2, is an essential component found in all synaptic vesicles (Feany *et al*, 1992). In addition to a proposed role in the transport of neurotransmitters into synaptic vesicles (Feany *et al*, 1992), SV2 has been linked to several mediators of $Ca^{2+}$ signaling which regulate and facilitate synaptic vesicle fusion events. SV2 is required for appropriate endocytic sorting of the $Ca^{2+}$ sensor synaptotagmin 1 to synaptic vesicles (Kaempf *et al*, 2015), as well as appropriate voltage-dependent $Ca^{+2}$ channel function at synapses (Vogl *et al*, 2015). A complicated story arises when SV2 levels are reduced, as different strategies have reported varying implications for synaptic transmission. In a mouse model of SV2 knockdown, two research groups have reported that responses to low frequency stimulation remained normal (Crowder *et al*, 1999; Janz *et al*, 1999). However, upon high-frequency successive stimulation, cultured neurons from these animals displayed sustained increases in $Ca^{2+}$-dependent synaptic transmission, which was suggested to explain an observed epileptic behavioral phenotype (Janz *et al*, 1999). In contrast, a third group has reported a significant reduction in amplitude of single evoked potentials in cultured hippocampal neurons from these animals (Custer *et al*, 2006). They also went on to demonstrate a reduction in the readily releasable pool of synaptic vesicles, as well as a reduced release probability (Custer *et al*, 2006). Genetic deletion of SV2 also leads to a reduction in spontaneous inhibitory potentials generated from GABAergic interneurons (Crowder *et al*, 1999). Taken together with our findings, particularly the results demonstrating the restoration of synaptic function and viability of GA-expressing neurons following SV2 replenishment, the loss of SV2 we observe in GA-expressing neurons is a prime candidate for the prevention of synaptic unloading in our ACSF perfusion stimulation paradigm. *In vivo*, a gradual reduction of properly released vesicles from cortical and motor neurons over time would have a strong impact on responses generated in recipient postsynaptic cells. Such a reduction in neurotransmitter inputs may lead to further improper signaling throughout the cortex, spinal cord, and at neuromuscular junctions. While we do not

suspect or propose that the loss of SV2 is the root cause of ALS in these patients, it should be considered as a further contributing factor to the multifactorial abnormal environment created by the presence of the C9orf72 hexanucleotide repeat expansion, associated RNA foci, and the other dipeptide repeat protein variants.

Intracellular $Ca^{2+}$ levels are tightly regulated in all cells and particularly in neurons (Zundorf & Reiser, 2011). Dysregulation of internal $Ca^{2+}$ has dramatic consequences for many fundamental cellular functions and consequently has been implicated in a variety of neurodegenerative conditions (Zundorf & Reiser, 2011). For example, many of the Alzheimer disease linked mutations and the toxic amyloid-beta fragment itself can each independently elevate levels of internal $Ca^{2+}$ (Chakraborty & Stutzmann, 2011). It is now thought that $Ca^{2+}$ dysregulation may be one of the early triggering events in AD pathogenesis which then feeds-forward into pathological signaling cascades (Chakraborty & Stutzmann, 2011). When looking at $Ca^{2+}$ influx in response to a stimulus designed to evoke neuronal firing, we found that cortical and motor neurons containing GA dipeptides displayed a significantly more robust response when compared with control cells. While we have not yet pinpointed the mechanism, we have considered several possibilities leading to this outcome. GA aggregates could be directly impacting cellular membrane permeability, or indirectly altering permeability through interactions with $Ca^{2+}$ channels. An alternative possibility is that this is a feed-forward mechanism of the cell to address the loss of synaptic vesicle component proteins. As $Ca^{2+}$ is required for effective synaptic vesicle fusion (Schneggenburger & Rosenmund, 2015), if the synaptic vesicles release is impaired, higher $Ca^{2+}$ influx could constitute a feed-back compensatory mechanism that neurons have in place to raise the probability of successful release of those vesicles. Finally, as was outlined above, perturbation of $Ca^{2+}$ levels could be a direct side effect of the altered levels of SV2 resulting from GA aggregates in neurites. Diminished SV2 interaction with $Ca^{2+}$ transport proteins and $Ca^{2+}$ sensors required for synaptic vesicle fusion could have dramatic implications for $Ca^{2+}$ entry and signaling at the synapse. While we have provided evidence that restoring SV2 levels alleviates increased $Ca^{2+}$ influx upon neuronal stimulation, we have not yet isolated the mechanism by which these fluctuations of SV2 levels contribute to our observations regarding cellular $Ca^{2+}$ entry in our model. Restoration of $Ca^{2+}$ homeostasis has been proposed as a therapeutic strategy in neurodegeneration, and our data suggest that patients with C9orf72-ALS/FTD may also benefit from recovery of cellular $Ca^{2+}$ regulation. Determining how and why this is occurring will be a valuable insight into possible therapeutic interventions for GA-mediated synaptic transmission deficits.

Concerns have commonly been raised as to the relevance of DPRs in terms of mechanistic contribution to C9orf72-ALS/FTD, due to the abundance and widespread DPR distribution throughout brain and spinal cord prior to neuronal cell death (Mackenzie *et al*, 2013, 2015; Gomez-Deza *et al*, 2015). Our findings here suggest that aggregates containing GA can be found within neuritic regions and affect synaptic transmission and synaptic vesicle composition. Our data indicate that by 48 hours, neurons expressing GA peptides display this aberrant elevation of $Ca^{2+}$ influx, reduction in SV2 levels, and abrogation of synaptic unloading. In future studies, detailing the sequence in which

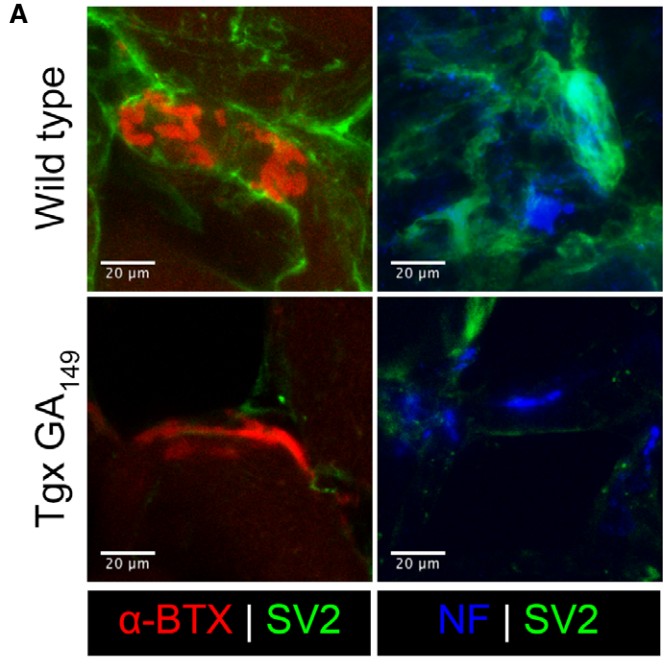

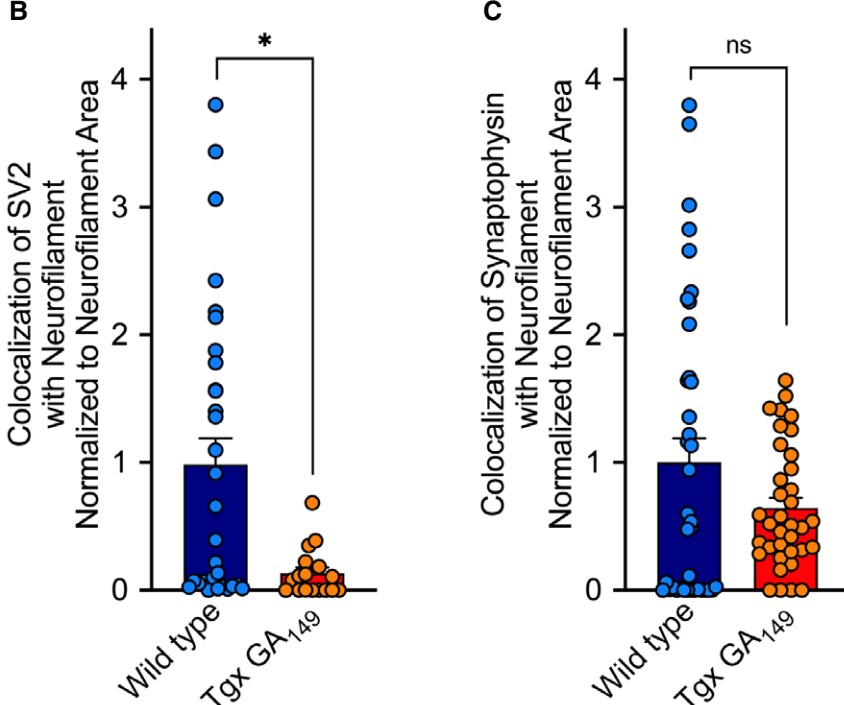

**Figure 10. SV2 is specifically reduced at neuromuscular junctions of symptomatic transgenic GA$_{149}$-CFP mice.**

Muscle sections (30 μm) were cut from snap-frozen tissue of 20-month-old GA$_{149}$ mice (Tgx GA$_{149}$) or wild-type controls. Immunohistochemical staining was performed to assess synaptic protein levels in proximity to NMJs. Sections were immunostained for SV2 or synaptophysin and counterstained with neurofilament.

A    Left Panels: Representative images show SV2 levels (green) in proximity to α-bungarotoxin-positive NMJs (red) in wild-type animals (top) versus GA$_{149}$ animals (bottom). Right panels: Representative images show SV2 levels (green) overlapping with neurofilament (blue) at synaptic terminals approaching muscle in wild-type animals (top) versus GA$_{149}$ animals (bottom).

B, C    Quantification of SV2 (B) or synaptophysin (C) colocalized with neurofilament and normalized to neurofilament area. This analysis revealed a significant reduction in SV2 levels in GA$_{149}$ animals compared with wild types, while synaptophysin remained unaltered. Data presented as mean ± SEM. Unpaired $t$-test, *$P < 0.05$. A total of 10 non-overlapping regions of muscle from each of five animals per group were evaluated. Exact $P$-values can be found in Appendix Table S1.

these events occur will identify the mechanism involved and provide insight for a targeted approach for preventing the resultant downstream synaptic dysfunction. We suspect that disruption of functional synaptic signaling prior to overt loss of motor neurons is the culprit for the behavioral phenotypes observed in our $GA_{149}$ mouse model. In multiple instances from human C9orf72 patient cases, staining of postmortem tissue samples has revealed neurons containing GA dipeptide aggregates with dystrophic neurites (Vatsavayai *et al*, 2016; Sakae *et al*, 2018). A recent article further solidifies the potential pathogenic nature of GA and therapeutic value of targeting this dipeptide in C9orf72 ALS/FTD. In this study, administration of a GA targeting antibody in a mouse model of C9orf72-ALS/FTD, successfully attenuated neurodegeneration, improved behavioral outcomes, and increased survival (Nguyen *et al*, 2020). In human patients, we cannot disentangle the effects of the various C9orf72 proposed pathogenic mechanisms, but along with this very recently published work, our *in vitro* and *in vivo* studies certainly lend further credibility to the idea that GA may have synaptic consequences which can be therapeutically targeted.

Overall, we have made some headway in understanding how the presence of GA DPR aggregates may lead to behavioral motor deficits preceding death of motor neurons in the spinal cord. Additionally, these findings help better clarify the previously conflicting reports on whether GA is toxic to neurons (May *et al*, 2014; Wen *et al*, 2014; Zhang *et al*, 2014). We propose that progressively aberrant synaptic transmission over time as a result of GA neuritic inclusions leads to a very gradual toxicity of neurons. This concept would also begin to explain how there is such widespread distribution of DPRs and particularly of GA in brain and spinal cord of patients of C9orf72-ALS/FTD without a correlation to disease severity or to regions where neuronal dropout is most severely affected (Mackenzie *et al*, 2013; Davidson *et al*, 2014, 2016; Gendron *et al*, 2015; Gomez-Deza *et al*, 2015; Schludi *et al*, 2015). It has also recently been established that DPRs including GA can be transmitted from cell to cell (Chang *et al*, 2016; Westergard *et al*, 2016). Here, we have shown that GA inclusions are present and mobile within neurites at time points greatly preceding cell death, which would allow ample time for cell-to-cell transmission events to take place. *In vivo,* a consequence of such long-term cell-to-cell spread would be a pervasive area of affected tissue, which will slowly diminish in synaptic transmission capacity over time.

# Materials and Methods

## Reagents

### Antibodies

Abcam: anti-MAP2 (Abcam Cat# ab5392, RRID:AB_2138153), anti-Neurofilament Heavy (Abcam Cat# ab8135, RRID:AB_306298), anti-PSD-95 (Abcam Cat# ab18258, RRID:AB_444362), anti-SV2 (Abcam Cat# ab32942, RRID:AB_778192); Covance: anti-SMI32 (Covance Research Products Inc Cat# SMI-32R-100, RRID:AB_509997); DSHB: anti-SV2 (DSHB Cat# SV2, RRID:AB_2315387); GeneTex: anti-p62/SQSTM-1 (GeneTex Cat# GTX100685, RRID:AB_2038029); Millipore: anti-ChAT (Millipore Cat# AB144P, RRID:AB_2079751); Thermofisher: anti-Synaptophysin (Thermo Fisher Scientific Cat# MA5-16402, RRID:AB_2537921).

## Constructs

eGFP-$GA_n$ constructs used were previously characterized in Ref. Wen *et al* (2014). mCherry-$GA_n$ constructs were subcloned from these vectors. pGP-CMV-GCaMP6f was a gift from Dr. Douglas Kim (Addgene plasmid #40755; Chen *et al*, 2013). rSV2a-eGFP-pRRL was a generous gift from Dr. Sandra Bajjalieh.

## Other reagents

Agilent: Dako target antigen retrieval solution; AmericanBio: non-fat dry milk omniblock; Bio-Rad Laboratories: 4–20% Mini-PROTEAN TGX stain-free gels, broad-spectrum molecular weight ladder, Tris/glycine running buffer, Tris/glycine transfer buffer; Decon Laboratories Inc: ethanol; Electron Microscopy Sciences: 16% paraformaldehyde, Citifluor AF3; LI-COR Biosciences: IRDye 680 RD donkey anti-rabbit IgG, IRDye 680RD goat anti-mouse IgG, IRDye 800CW donkey anti-mouse IgG, IRDye 800CW goat anti-rabbit IgG; Millipore Sigma: β-mercaptoethanol, bovine serum albumin (BSA), Bradford reagent, calcium chloride dihydrate, deoxycholic acid, dithiothreitol (DTT), (ethylenedinitrilo)tetraacetic acid (EDTA), D-glucose, HEPES, hydrogen peroxide, immobilon-FL membranes, poly-D-lysine, magnesium chloride hexahydrate, OptiPrep density gradient medium, sodium dodecyl sulfate, sodium phosphate; National Diagnostics: Histoclear; Takara Bio USA: Lenti-X Packaging Single Shots, Lenti-X Concentrator, Lenti-X Gostix; Thermofisher: AlexaFluor546-conjugated anti-mouse or anti-rabbit secondary antibodies, AlexaFluor647-conjugated anti-mouse or anti-rabbit secondary antibodies, α-bungarotoxin AlexaFluor594 conjugate, B27 supplement, diamidino-2-phenylindole (DAPI), FM4-64 fluorescent dye, glutamine, heat-inactivated horse serum, Hoechst 33258 pentahydrate (bis-benzimide), laminin, Lipofectamine 2000 reagent, LysoTracker Deep Red, methanol, MitoTracker Deep Red FM dye, neurobasal medium, neutravidin protein, normal goat serum, phosphate-buffered saline (PBS), potassium chloride, protease inhibitors, sodium bicarbonate, sodium chloride, SYTO RNASelect Green Fluorescent cell stain, Tris–HCl, Triton X-100, trypsin, Tween-20; US Biological: Nonidet (NP-40); Vector Labs: Vectashield Antifade mounting media with DAPI.

## Primary rat cortical and motor neuron cultures

All experiments were performed in accordance with the guidelines set forth by the Institutional Animal Care and Use Committees at Thomas Jefferson University. Primary rat cortical neurons were isolated from embryonic day 18 rats as previously described (Kayser *et al*, 2006; Hanamura *et al*, 2017). Timed-pregnant female Sprague Dawley rats were procured from Charles River and delivered on the day of embryonic dissection. Immediate transfer to the laboratory occurred upon arrival. Dissociated cortical neural cell suspensions were plated on poly-D-lysine-coated coverslips or 35-mm dishes with glass-bottom centers in Neurobasal medium with B27 supplement at 37°C with 5% $CO_2$. Primary rat motor neurons were prepared from embryonic day 14.5 spinal cords with slight modifications to the established protocol (Magrane *et al*, 2012). The following morning, embryonic spinal cords were dissected and mechanically fragmented into small pieces, then incubated with 0.025% trypsin. Following manual trituration, cells were passed through a 4% (w/v) bovine serum albumin (BSA) cushion and then centrifuged through 10.4% (v/v) Optiprep cushion (Sigma) for 55 min without brake. Collected motor neuron bands were then

spun through 2 sequential 4% (w/v) BSA cushions. Cells were resuspended in Neurobasal medium with B27 supplement, glutamine (0.25%), β-mercaptoethanol (0.1%), and horse serum (2%). Dissociated motor neurons were plated on poly-D-lysine and laminin-coated coverslips or 35-mm dishes with glass-bottom centers and maintained at 37°C with 5% $CO_2$. Animals were housed in the Bluemle Life Sciences Animal Facility following our IACUC-approved protocol in standard cages with *ad libitum* access to food and water.

### Neuronal transfections

Transfection of primary cortical and motor neurons with plasmids encoding GA at different lengths or control eGFP-FLAG were performed as previously reported (Wen *et al*, 2014). For GCaMP experiments requiring use of the 488 FITC channel to assess $Ca^{2+}$ levels, the eGFP tag was switched for mCherry using standard subcloning techniques. In order to represent functionally mature neurons, we began all experiments at time points where primary neuron cultures were capable of excitatory synaptic communication. By DIV7, cortical neuron cultures derived from rat express a full complement of NMDA and AMPA receptor subunits (Janssens & Lesage, 2001), and by DIV9, the time point we have examined for neuronal function is functionally competent for synaptic transmission (Virdee *et al*, 2017). Similarly, primary rat motor neurons have been shown to undergo robust glutamate-induced excitotoxicity at DIV7, underlying our rationale for beginning motor neuron experiments at DIV5 and assessment at DIV7 (Vincent *et al*, 2004). Transfection of DIV7 cortical neurons and DIV5 motor neurons was accomplished through the use of Lipofectamine 2000 reagent and 1 μg total DNA/well. When co-transfection was necessary, Td-Tomato or GCaMP6f was introduced at a 1:4 cDNA ratio with the desired eGFP or mCherry GA constructs, respectively.

### i3 cortical neuron cultures

Cortical neurons derived from iPSCs were produced using transcription factor-mediated differentiation using a technique dubbed i3 neurons (integrated, inducible, and isogenic). Two cell lines were utilized, which were derived from a patient with a C9orf72 repeat expansion, and an isogenic control in which the repeats were removed from the patient line. These were kind gifts from Dr. Michael E. Ward and Dr. Kevin Talbot. Prior to tissue donation and generation of pluripotent stem cells, informed consent was obtained from the C9orf72 ALS patient. The tissue collection conformed to the principles set out in the WMA Declaration of Helsinki and the Department of Health and Human Services Belmont Report. In the iPS stage, cells were maintained in Essential 8 Flex media (Gibco) on matrigel-coated plates (Corning). Differentiation was performed using the techniques outlined in Fernandopulle *et al* (2018) in which the master neuronal transcriptional regulator neurogenin-2 (NGN2) was stably integrated into a safe harbor locus in the iPS lines under the control of a doxycycline-inducible promoter. To differentiate iPS cells into cortical neurons, cells were dissociated into single cells and plated on matrigel-coated plates in induction media (DMEM/F12, N2, NEAA, and L-glutamine) containing 2 mg/ml doxycycline. Cells were treated with doxycycline for 3 days before being dissociated and plated on PLO-coated plates. Cells were then allowed to mature

for an additional 18 days in cortical maturation media consisting of BrainPhys media (STEMCELL) supplemented with BDNF, NT-3, and Laminin, with half media changes twice weekly. Cells were either harvested or fixed 21 days post-differentiation.

### iMN lysates

iMNs cell pellets were a generous gift from the Answer ALS Consortium. iMNs were generated from control and C9orf72-ALS patients and were grown and differentiated according to the protocol set forth by Neurolincs (http://neurolincs.org/technologies/). Prior to tissue donation and generation of pluripotent stem cells, informed consent was obtained from all consortium participants. Tissue collection protocols conformed to the principles set out in the WMA Declaration of Helsinki and the Department of Health and Human Services Belmont Report. The spinal neuron protocol used reproducibly generates a mixed neuronal population including 20–30% Islet1-positive neurons. Following 32 days of differentiation, iMNs were rinsed with PBS, scraped with a cell scraper, and pelleted by centrifugation at 1,000 *g* for 5 min. The supernatant was aspirated; pellets were flash-frozen in liquid nitrogen and were stored at −80°C. In our hands, these cells were resuspended in RIPA buffer, and immunoblotting was performed as described.

### Immunocytochemistry

Primary neuron cultures on glass coverslips were fixed in 4% paraformaldehyde for 8 min. Cells were permeabilized and blocked in phosphate-buffered saline (PBS) with 5% BSA and 0.1% Triton X-100 for 1 h at room temperature, incubated with primary antibodies at 4°C overnight, and secondary antibodies at room temperature for 1 h. Coverslips were affixed to slides in Vectashield Antifade mounting media with DAPI (Vector Labs) to stain nuclei. Three PBS washes occurred between each step. The following primary antibodies were used: anti-SMI32 (Covance Research Products Inc Cat# SMI-32R-100, RRID:AB_509997, 1:1,000), anti-MAP2 (Abcam Cat# ab5392, RRID:AB_2138153, 1:5,000), anti-Neurofilament Heavy (Abcam Cat# ab8135, RRID:AB_306298, 1:1,000), anti-SV2 (DSHB Cat# SV2, RRID:AB_2315387, 1:200), anti-Synaptophysin (Invitrogen SP11, Thermo Fisher Scientific Cat# MA5-16402, RRID: AB_2537921, 1:200), anti-PSD-95 (Abcam Cat# ab18258, RRID: AB_444362, 1:1,000), and anti-p62/SQSTM-1 (GeneTex Cat# GTX100685, RRID:AB_2038029, 1:1,000). Secondary antibodies used at a 1:500 dilution were AlexaFluor546 or AlexaFluor647-conjugated. Confocal microscopy was used to capture images (Olympus FV1000). For each field of view, 10 images were acquired with a 1μm step size, which were projected into a single *z*-stack image. Ten non-overlapping fields per coverslip were generated for experiments requiring quantitative analysis. For all microscopy-based imaging, laser and filter settings for each experiment were calibrated according to internal experimental control groups (i.e., eGFP or mCherry-expressing cells), to account for biological variability and ensure reproducibility.

### Assessment of synaptic protein puncta and neurite length

$GA_n$ was transfected into neurons. At 48 h post-transfection, neurons were immunostained for synaptic component proteins SV2,

synaptophysin, or PSD-95, counterstained with neurofilament and DAPI, and imaged by confocal microscopy. Quantification of puncta for each of the aforementioned synaptic components along neurites in eGFP$^+$ cells was accomplished using NIH ImageJ software (V1.50i, Bethesda, MD). Puncta were assigned on the z-stack projection images based visually on localization within the neurofilament area and distinct borders separating them from adjacent puncta. To determine total neurite length, images were analyzed using the NeuronJ plug-in feature of ImageJ. In eGFP$^+$ cells, each neurite was individually traced, with the sum total of length per cell used for analysis.

## Live-cell aggregate, mitochondrial, lysosome, and total RNA trafficking

Neurons underwent live-cell imaging 48 h after GA$_n$ expression. A high-resolution confocal microscope was used to rapidly capture images to assess individual GA aggregate, mitochondria, lysosome, and RNA velocities. GA$_n$ aggregates were observed through the use of the 488 nm laser. MitoTracker Deep Red FM dye (Molecular Probes, 50 nM) labeled mitochondria. This cell-permeable compound has an excitation maximum at 644 nm, enabling visualization with a 637 nm laser. Brightfield images were taken to define neurite tracts. Images were taken at 2-s intervals for 5 min. Lyso-Tracker Deep Red (Molecular Probes, 50 nM) and SYTO RNASelect Green Fluorescent cell Stain (Molecular Probes, 500 nM) instead label lysosomes and total RNA respectively but operate under the same principles as MitoTracker, using the same imaging parameters. LysoTracker has an excitation maximum of 647 nm, still allowing visualization with the 637 laser, while SYTO RNASelect has a peak excitation at 490 nm, switching usage to the 488 laser, and requiring our transfections to be performed with our mCherry-based GA constructs. Individual velocities for GA$_n$ aggregates, mitochondria, lysosomes, and total RNA were determined using ImageJ software through features of the kymograph analysis function.

## SV2 mRNA Beacon

A target probe specific for SV2 mRNA was obtained through Bio-Synthesis, Inc. with the sequence 5′[Cy5]GAACCAGGTGGTGACTGTTTA[BHQ-2]3′. This was resuspended at 100 μM in TE buffer (pH 8.0) and aliquoted for use. The day prior to transfection, 2.7 μl of neutravidin solution was added to a 9 μl volume of beacon and incubated for 1.5 h at room temperature shielded from light and then moved to 4°C overnight. This solution was then added along with the normal transfection components for eGFP or eGFP-GA50 along with a co-transfected td-Tomato cell-filling plasmid. Following 2 days of expression, confocal imaging was used to assess beacon and GA cellular localization.

## Synaptic vesicle release

Vesicle exocytosis was measured through application of FM4-64 fluorescent dye (Invitrogen), using TRITC imaging parameters (Gaffield & Betz, 2006; Verstreken *et al*, 2008). After 48 h of eGFP-GA$_n$ expression, neurons were incubated with 10 μM FM4-64 dye in high K$^+$ ACSF, followed by a wash in low-K$^+$ ACSF to regain baseline. High K$^+$ ACSF (ACSF Stim: 95 mM NaCl, 50 mM

KCl, 5 mM sodium bicarbonate, 1.2 mM sodium phosphate, 1 mM MgCl$_2$ hexahydrate, 10 mM glucose, 10 mM HEPES, and 1.33 mM CaCl$_2$ dihydrate) promotes neuronal firing and uptake of dye through vesicle recycling. Low-K$^+$ ACSF (140 mM NaCl, 5 mM KCl, 5 mM sodium bicarbonate, 1.2 mM sodium phosphate, 1 mM MgCl$_2$ hexahydrate, 10 mM glucose, and 10 mM HEPES) allows for cell recovery and normalization to baseline. Using our Nikon Eclipse T$i$ microscope, eGFP$^+$ cells were identified, and then, baseline measurements were acquired at 40× magnification on the TRITC channel for 3 min, with images captured every second. Cells were stimulated by perfusion with ACSF Stim, with images captured every second for 2 min. Signal normalization to average baseline recording accounted for any initial variability in dye loading. Ten FM4-64 puncta regions were selected from each eGFP$^+$ neuron for analysis using NIS-Elements Imaging software. Kinetics are presented as change in fluorescence (ΔF) normalized to basal fluorescence (F), ΔF/F, as an indicator of stimulated dye release.

## Neuronal Ca$^{2+}$ imaging

Neurons were co-transfected with mCherry-GA$_n$ constructs and pGP-CMV-GCaMP6f (a gift from Douglas Kim; Addgene plasmid #40755; Chen *et al*, 2013). After 48 h, mCherry$^+$ cells were identified and cells were switched to a low potassium artificial cerebrospinal fluid (ACSF Low-K$^+$) without pH indicator dyes for live-imaging purposes. Basal GCaMP fluorescence intensity (FITC channel) was monitored for 3 min prior to induced depolarization of neurons with ACSF Stim with or without CaCl$_2$. The contribution of Ca$^{2+}$ influx was determined by stimulation in ACSF containing high KCl but no Ca$^{2+}$. Images were captured at 40× magnification at 250 ms intervals. Imaging proceeded for an additional 3 min post-initiation of ACSF perfusion. Using NIS-Elements Imaging software, ROIs of the cell soma were produced and fluorescence values generated across the imaging session. Following baseline normalization, the peak change in fluorescence ΔF/F value was recorded for each mCherry$^+$ neuronal cell body.

## Longitudinal live-cell imaging survival analysis

Cortical and motor neurons co-transfected with eGFP-GA$_n$ and td-Tomato plasmids underwent longitudinal live-cell imaging as previously described to perform survival and risk of death analyses for poly-GA tracts spanning up to 400 repeats (Wen *et al*, 2014). With our fully automated microscopy system utilizing a Nikon Eclipse Ti microscope and CoolSNAP ES2 high-performance CCD camera, stage and shutter movements, focusing, and image acquisition are all controlled by NIS-Elements Imaging software (Nikon). This system allows for hundreds of neurons from each group to be imaged at 24-h intervals. Td-tomato fluorescence was used as a marker of cell survival, as loss of fluorescence stemming from plasma membrane disruption indicates cell death. eGFP and Td-tomato signals were monitored for 13 days in cortical neurons and 10 days in motor neurons. Visual inspection of files generated from the automated imaging protocol enabled assessment of survival for individual neurons over time, with death defined as the day on which a neuron lost td-Tomato fluorescence.

## Quantitative reverse transcription–polymerase chain reaction (qRT–PCR)

The expression of SV2 in i3 cortical neuron cultures 21 days post-differentiation and cortical neurons following 4 days of SV2 upregulation via rSV2a-eGFP-pRRL lentiviral expression was quantified using quantitative reverse transcription–polymerase chain reaction (qRT–PCR). mRNA was extracted with Trizol, and 500 ng of RNA was converted to cDNA by the Invitrogen Superscript First-strand kit. Quantitative PCR was performed using PowerUp SYBR Green. Samples were measured in triplicate for each experiment from three independently differentiated wells of each genotype for the i$^3$ cultures, and three independently treated wells for the cortical neuron groups. Data were normalized using GAPDH levels and analyzed according to the $\Delta\Delta C_T$ method. Primer pairs obtained from Integrated DNA Technologies (Coralville, IA) for each gene are listed. Primers for i$^3$ human: SV2 forward: CTG CTC CCA GTG ATG GTT ATT, SV2 reverse: GAT CTC ATC ATC CTC GTC ATG G. Primers for rat cortical neurons: SV2 forward: GAA CCA GGT GGT GAC TGT TTA, SV2 reverse: TCC AGC AAC GGA ATG ATG AG. Primers for GAPDH can be used cross-species and were used for both experiment groups. GAPDH forward: AGG TCG GTG TGA ACG GAT TTG, GAPDH reverse: TGT AGA CCA TGT AGT TGA GGT CA.

## Production of rSV2a-eGFP-pRRL lentivirus

We obtained the rSV2a-eGFP-pRRL lentiviral construct as a generous gift from the Bajjalieh laboratory at the University of Washington, which produces rat SV2 coupled with an eGFP tag (Schivell *et al*, 1996, Yao *et al*, 2010). Using the manufacturer's protocol, we generated lentiviral particles in-house by Lenti-X Packaging Single Shots and concentration with Lenti-X Concentrator (Takara Bio). Lentiviral GoStix determined effective viral production. 1 μl of virus was added per 500 μl of cell culture medium. Following 4 days of expression in mature cortical neurons, effective SV2 upregulation of lentiviral-transduced neurons compared to untreated neurons was determined following qPCR and immunoblot analysis. This effective virus was then used at the designated volume-based ratio for all rescue experiments. Virus was always added immediately following the completion of the transfection protocol to express GA constructs.

## Immunoblotting and immunohistochemistry of mouse tissue

All animal experimental protocols were conducted in accordance with the European and German national guidelines for the use of animals and were approved by the governmental committee (Regierungspräsidium Oberbayern, Germany).

### Immunoblotting

Fresh-frozen spinal cords were prepared from 20-month-old GA$_{149}$-CFP and wild-type mice in the C57BL/6N background, which have been previously validated and characterized (Schludi *et al*, 2017). Mice were housed in standard cages, in a pathogen-free facility with a 12-h night/day light cycle. Animals were given *ad libitum* access to food and water. All experiments were performed in accordance with the guidelines set forth by the Institutional Animal Care and Use Committees at Thomas Jefferson University. Dissociated single-cell suspensions were prepared by homogenization in buffer containing:

10 mM HEPES pH 7.9, 1.5 mM MgCl$_2$, 10 mM KCl, 0.5 mM DTT, and 1:100 protease inhibitor cocktail, followed by centrifugation at 845 $g$ at 4°C for 5 min. Cells were resuspended in RIPA buffer (150 mM NaCl, 1% Nonidet (NP-40), 0.5% deoxycholic acid, 0.1% sodium dodecyl sulfate, 50 mM Tris–HCl, 2 mM EDTA, and 1:100 protease inhibitor cocktail), pulse-sonicated 3 times for 10 s each with 2 min rest on ice between rounds, and spun at 21,130 $g$ at 4°C for 30 min. Protein concentrations of supernatant fractions were determined by the Bradford method. 5 μg of protein was loaded onto 4–20% Mini-PROTEAN TGX stain-free gels for electrophoretic separation, along with a broad-spectrum molecular weight ladder. After separation, total protein levels were assessed by a Bio-Rad ChemiDoc Touch Imaging System using the pre-set stain-free gel parameters. Proteins were transferred onto Immobilon-FL membranes (Millipore) and blocked in PBS with 0.1% Tween-20 (PBST) and 5% milk for 30 min at room temperature. Membranes were incubated overnight at 4°C with primary antibodies in PBST + 5% milk. The following primary antibodies were used: anti-SV2 (Abcam ab32942, Abcam Cat# ab32942, RRID:AB_778192, 1:1,000), anti-synaptophysin (Thermo Fisher Scientific Cat# MA5-16402, RRID:AB_2537921, 1:1,000), and anti-PSD-95 (Abcam Cat# ab18258, RRID:AB_444362, 1:1,000). Following PBST washes, membranes were incubated with antigen-specific fluorescent probe-conjugated secondary antibodies (1:15,000 dilution LiCor secondaries) in PBST + 5% milk for 1 h at room temperature. Membranes were visualized using an Odyssey Infrared Imaging System (LiCor) and ImageStudio software. Using ImageJ, bands of interest were specified at the heights corresponding to the molecular weight of each protein of interest. These selections were analyzed to determine pixel intensities of each lane. To normalize results, a horizontal section of the Bio-Rad stain-free gel total protein image spanning a broad molecular weight range was selected. Overall intensity was determined for each lane, allowing for normalization without bias toward a single representative loading control and taking into account variability in tissue lysate extracts. Four animals from both poly-GA$_{149}$-CFP and wild-type animal cohorts were evaluated. For each of these cohorts, two animals were male and two were female.

### Immunohistochemistry

Following euthanasia in accordance with approved protocols, 20-month-old mice were transcardially perfused with 1% sterile PBS. Dissected spinal cord tissue was formalin-fixed for 2 days and paraffin-embedded. Staining was performed on glass slides containing 10-μm-thick sections cut from cervical and upper thoracic regions of these tissue samples. Slides were heated to 55°C overnight, deparaffinized in Histoclear, and rehydrated in sequential 100, 95, 90, and 70% ethanol washes. Endogenous peroxidase activity was blocked by incubation in 3% hydrogen peroxide in methanol for 30 min. Antigen unmasking was achieved through incubation in citrate target antigen retrieval solution at 95°C for 1 h. Sections were blocked in 10% normal goat serum in PBS for 1 hour at room temperature. Slides were incubated overnight at 4°C with primary antibodies against ChAT (Millipore Cat# AB144P, RRID:AB_2079751, 1:1,000) and secondary labeling with AlexaFluor546 for motor neuron-specific staining. Hoechst was used to stain nuclei. Slides were mounted in Citifluor AF3, and a Nikon A1$^+$ microscope with NIS-Elements Imaging software was used for image capture and processing. Motor neuron counts were achieved by a manual count of motor neurons/anterior horn using ImageJ software. Number of

## The paper explained

### Problem

A hexanucleotide repeat expansion in the C9orf72 gene is the most common cause of familial and apparently sporadic cases of ALS. While evidence has shown that loss-of-function haploinsufficiency of C9orf72 protein levels may lead to immunological problems, gain-of-function toxic effects caused by accumulation of RNA foci and aggregates of dipeptide repeat proteins (DPRs) both resulting from aberrantly expanded RNA transcripts are the likely pathogenic culprits behind the death of motor neurons. As many C9orf72 patients fall into the sporadic ALS cohort, they possess a high DPR burden by the time of diagnosis. The most abundant of these DPRs is the GA dipeptide, which invokes very gradual cellular toxicity.

### Results

Here, we show that mobile GA aggregates are present in the cytoplasm and axons of cortical and motor neurons. In functional studies, we found that expression of GA resulted in increased $Ca^{2+}$ influx but reduced synaptic unloading in response to a stimulus for neuronal firing. Examination of synaptic proteins revealed that synaptic vesicle-associated protein 2 (SV2) was specifically reduced. While neurons containing GA ultimately succumbed to toxicity, synaptic protein and transmission deficits occurred several days prior to this demise. Importantly, if we restore SV2 levels in GA-expressing cells, we are able to successfully recover synaptic function and prevent neuronal death. We validated reduction of SV2 in 20-month-old GA transgenic mice, without loss of motor neurons. Finally, we confirmed specific loss of SV2 in C9orf72-ALS patient-derived induced cortical and motor neurons.

### Impact

Understanding and targeting the cellular consequences evoked by the abundant and gradually toxic DPR, GA may be a promising therapeutic option for C9orf72-ALS patients. In isolating molecular mechanisms of synaptic deficits induced by GA, we provide the groundwork for future identification of intervention therapies specifically designed to maintain the synapses required for motor neuron-muscle connectivity and therefore prolong muscle strength in C9orf72-ALS patients.

motor neurons per anterior horn for each experimental animal was generated using bilateral anterior horn images and counting both ChAT$^+$ cells and appropriately sized and shaped motor neuron Hoechst nuclei. Four 20-month-old animals from both GA$_{149}$-CFP and wild-type cohorts were evaluated. Hindlimbs were snap-frozen in liquid nitrogen and stored at −80°C. Staining of NMJs was performed on glass slides containing 30 µM thick sections cut from these samples. Slides were heated to 55°C overnight, deparaffinized in Histoclear, and rehydrated in sequential 100, 95, 90, and 70% ethanol washes. Sections were blocked in 10% normal goat serum in PBS for 1 h at room temperature. Slides were incubated overnight at 4°C with primary antibodies against anti-SV2 (DSHB Cat# SV2, RRID:AB_2315387, 1:200) or anti-Synaptophysin (Invitrogen SP11, Thermo Fisher Scientific Cat# MA5-16402, RRID:AB_2537921, 1:200) and anti-Neurofilament Heavy (Abcam Cat# ab8135, RRID:AB_306298, 1:1,000), and labeling with AlexaFluor secondaries. Hoechst was used to stain nuclei; α-bungarotoxin AlexaFluor594 conjugate was used to label neuromuscular junctions. Slides were mounted and imaged as was performed for spinal cord sections. Five 20-month-old animals from both GA$_{149}$-CFP and wild-type cohorts were evaluated, with the individual performing analysis blinded to genotype. For each of these cohorts, two animals were male and three were female.

## Statistical analysis

All statistical analyses were performed using Prism 8.0 software (GraphPad). Differences between Kaplan–Meier survival curves were assessed by the log-rank test, with hazard ratios also performed to assess direct comparison between selected conditions. Statistical significance for pairwise assessment of each time point in synaptic unloading experiments was accomplished using the Sidak–Bonferroni method. One-way ANOVA with *post hoc* Dunnett's multiple comparison test, one-way ANOVA with uncorrected Fisher's LSD test, and unpaired *t*-tests with Welch's correction were performed where indicated. All data are expressed as mean ± SEM, with values of $P < 0.05$ considered significant.

## Data availability

No large scale data sets have been generated in this study.

**Expanded View** for this article is available online.

## Acknowledgements

We thank Dr. Dalva's laboratory for providing embryonic rat cortices for cortical neuron preparations. We thank Drs. Michael E. Ward and Kevin Talbot for sharing i3 iPS cell lines, Dr. Jeffrey D. Rothstein and the Answer ALS consortium for providing us with whole-cell pellets of induced MNs, and Dr. Sandra Bajjalieh for her generous sharing of the rSV2a-eGFP-pRRL lentiviral plasmid. We would also like to acknowledge members of the Jefferson Weinberg ALS Center for suggestions and critical evaluation of this work. This work was supported by funding from the NIH (RF1-AG057882-01 and R21-NS0103118 to D.T., R01-NS062047 to L.M., and R00 NS091486 to A.R.H.), the Muscular Dystrophy Association (D.T.), the Robert Packard Center for ALS Research (D.T.), the Family Strong 4 ALS foundation and the Farber Family Foundation (B.K.J. and P.P).

## Author contribution

Conceptualization: BKJ, DT; methodology: BKJ, KJM, KAR, AB, BMC, KK, LM; investigation: BKJ; KAR, AB, BMC; formal analysis: BKJ; visualization: BKJ, DT; writing—original draft: BKJ, DE, DT; funding acquisition: ARH, PP, DT; resources: MHS, TW, XW; and supervision: ARH, PP, DT.

## Conflict of interest

The authors declare that they have no conflict of interest.

## For more information

(i)  https://www.jefferson.edu/university/farber_institute/weinberg_als_center.html
(ii)  http://www.alscenter.org/
(iii)  http://www.alsa.org/
(iv)  https://www.answerals.org/
(v)  http://neurolincs.org/technologies/

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
