## [Review Process File · EMBO Molecular Medicine]

Synaptic dysfunction induced by glycine-alanine dipeptides in C9orf72-ALS/FTD is rescued by SV2 replenishment

Brigid K. Jensen, Martin H. Schuldi, Kevin J. McAvoy, Katelyn A. Russell, Ashley Boehringer, Bridget M. Curran, Karthik Krishnamurthy, Xinmei Wen, Thomas Westergard, Le Ma, Aaron R. Haeusler, Dieter Edbauer, Piera Pasinelli, Davide Trotti

Review timeline:

Submission date:	4th Apr 2019
Editorial Decision:	10th Apr 2019
Authors' correspondence:	10th Apr 2019
Editorial correspondence:	11th Apr 2019
Revision received:	5th Aug 2019
Editorial Decision:	18th Sep 2019
Revision received:	6th Feb 2020
Editorial Decision:	9th Mar 2020
Revision received:	23rd Mar 2020
Accepted:	26th Mar 2020

Editor: Céline Carret

Transaction Report:

1st Editorial Decision

10th Apr 2019

Thank you for submitting your article "Glycine-alanine dipeptides in C9orf72-ALS/FTD cause synaptic release deficits without neuronal death" to EMBO Molecular Medicine. I have now carefully read your manuscript and discussed it with my colleagues. I regret to say that we find the manuscript not well suited for publication in EMBO Molecular Medicine and therefore have decided not to proceed with peer review.

The manuscript reports on polyGA dipeptide repeats mechanism in primary neurons, cortical and motor, in C9orf72-ALS in vitro and in vivo in GA149 animals. SV2 is found down-regulated while calcium influx is increased. We appreciate that GA aggregates are found to move within axons and dendrites, affecting synaptic transmission before cell death occurs. However, while certainly relevant to better understand the underlying mechanistic process, unfortunately, the nature of the data does not fit well within our scope, as no translational nor clinical novel implications are drawn that could be exploited in the disease setting. As such, I am afraid that we can not offer further consideration to your article for EMBO Molecular Medicine.

In the interest of your manuscript and your time, I am providing you with an early editorial decision that will allow you to submit elsewhere without further delay. I am sorry to disappoint you on this occasion and I thank you once more for your interest in EMBO Molecular Medicine.

Authors' correspondence

10th Apr 2019

Thanks for considering our manuscript. We understand and agreed on your decision, as indeed our paper as submitted lacks novel translational and clinical implications.

To overcome this issue we are considering a gene therapy approach to rescue the downregulation of SV2 and restore the compromised phenotype. This therapeutic approach can be done in vitro on cortical and motor neurons alike by restoring SV2 levels using a lentiviral SV2-encoding construct and assessing the impact of the survival of GA-expressing neurons and their functional ability to perform synaptic vesicle release.

Indeed, if this approach works, we would have validated - at least in vitro - SV2 as a valuable therapeutic target and provide the clinical and translation implications that are currently lacking from our study. As an extension, a similar approach could be attempted for the GA149 mouse line generated by Dr. Edbauer. However, this line as been discontinued and currently is only maintained in cryopreservation. It would be unfeasible for us to do these translational studies in the mice at this time.

I was, therefore, wondering if by adding the in vitro translational approach experiments to the current study would allow our paper to be considered for publication in EMBO Mol.Med.

Editorial correspondence

11th April 2019

While the gene therapy approach is in principle of interest and would definitely increase the translational relevance, I am afraid that your proposal to provide an in vitro gene therapy approach would not be enough to really fit our scope.

I fully understand that the GA149 mice are not breeding and reviving the colony would take a long time, which seems to exclude that possibility.

Another possibility, that would be enough for us for at least sending the paper out, would be to use patients cells.

Authors' correspondence

1st Aug 2019

We were actually able to follow up with new experimental data in iPSC neurons derived from C9 patients on our original studies. As discussed in our email conversation below, I hope this new addition to our manuscript will make it suitable for editorial consideration. We are ready to submit it and were planning on doing it this coming Monday.

However, before submitting it I just wanted to check if your invitation was still open following our addition of new experiments.

In summary, we have specifically examined the contribution of GA dipeptides to neuronal function, and have identified a novel and potentially pathogenic mechanism by which GA may contribute to failure of functional synaptic connectivity in C9orf72-ALS. We have also established that these findings are relevant to C9orf72 ALS and FTD patients, as we have validated our findings in patient derived induced pluripotent stem cells (IPSCs) differentiated into both cortical and motor neurons. Notably, we have shown the following:

1. In cortical and motor neurons, mobile cytosolic GA aggregates can localize to dendrites and axons.
2. Functionally, expression of GA results in increased calcium influx but reduced synaptic unloading upon conditions of evoked neuronal firing.
3. Synaptic vesicle associated protein 2 (SV2) is specifically reduced in GA-containing neurons, amongst a panel of synaptic protein components examined. This was characterized in vitro and confirmed in vivo, in 20-month old transgenic animals which display behavioral motor abnormalities but have no loss of spinal cord motor neurons.
4. Specific loss of SV2 was confirmed in i3-cortical neurons derived from a C9orf72 repeat expansion patient compared with the isogenic control in which the repeat expansion was excised. Specific loss of Sv2 is also suggested in differentiated IPSC motor neurons derived from C9orf72-

ALS patients compared with healthy controls.

5. Calcium activity, reduction of SV2, and transmission deficits occur several days prior to when neurons ultimately succumb to GA-induced neurotoxicity in vitro, suggesting a potential window for therapeutic intervention.

1st Revision - authors' response

5th Aug 2019

Thank you for the standing invitation for the submission of our manuscript.

2nd Editorial Decision

18th Sep 2019

Thank you for the submission of your manuscript to EMBO Molecular Medicine. I apologise for the delay in getting back to you on your article. An unfortunate combination of holidays and travelling and a very busy office. We have now heard back from the three referees whom we asked to evaluate your manuscript.

You will see that while the topic of your article is of great interest, it looks like the data need to be substantiated and some mechanism should be provided. In addition, controls, statistical analyses, better imaging and additional experiments are requested to support the main findings, along with maybe additional iPSN lines as well. Although we would like to invite a revision of your article to be further evaluated, please be warned against returning an incomplete revision. Referees are aware that they are asking for a lot (and I am ready to give you more time should you need it), but they feel that it's needed to make the paper better and to the level we expect at EMBO Molecular Medicine.

We would therefore welcome the submission of a revised version within three months for further consideration and would like to encourage you to address all the criticisms raised as suggested to improve conclusiveness and clarity. Please note that EMBO Molecular Medicine strongly supports a single round of revision and that, as acceptance or rejection of the manuscript will depend on another round of review, your responses should be as complete as possible.

***** Reviewer's comments *****

Referee #1 (Remarks for Author):

Intronic G4C2 repeat expansion in C9orf72 is the most common cause of familial ALS/FTD. One of the pathogenic mechanism is the aberrant accumulation of five dipeptide repeat proteins produced by RAN translation. DPR inclusions have been detected in C9orf72 repeat expansion related ALS/FTD postmortem brain. Differ from the arginine-rich DPRs, namely GR and PR, GA aggregation is widely detected in brain and spinal cord causing more subtle cellular impairment, including neuronal loss, yet is still sufficient to trigger motor deficits. This phenomenon raises the possibility that GA expression might contribute to neuronal dysfunction without significant cell death. In the current paper, the authors utilized a serial of imaging methods showed that GA aggregates are mobile in the axon and dendrites of the neuron. Calcium influx and synaptic release are disrupted due to the presence of GA. Meanwhile, SV2 expression is down-regulated in multiple in vivo and in vitro models which the authors suggest contributing to the progression of the disease. While the paper raises an important question and provides some intriguing evidence that synaptic function is interrupted before neuronal death because of the presence of GA, there are some concerns and several data are not very convincing to support the conclusions. And there is no mechanism how GA aggregates lead to synapse dysfunction.

Major concerns:

- In Fig 1, the authors tried to show that GA50 aggregates are presented in the axon and dendrite of the primary cultured neuron. However, it is difficult to visualize the expression pattern of GA50 in

neurons with the overexposed MAP2 and neurofilament signal. Especially the day 4 and day 8 images are not convincing. The GFP signal looks like background noise, as similar signals also shows in the extracellular space. A split channel of eGFP-GA50 and dimming neuronal marker signal is necessary to show the overall pattern of GA50 aggregates in the cell. Higher resolution images are necessary.

- How about the percentage of GA aggregates in soma vs neurite? Any difference between cortical neurons and motor neurons?
- An eGFP only transfection should be used as a control to show that the transfection itself does not contribute to the aggregation of the protein.
- In Fig3A, it is hard to distinguish the real signal of FM4-64 dye from the background. What are the real signals at the synapse and what are extracellular signals? A bright field overlay or synapse labeling (such as by co-expressing with a synapse marker), as well as higher resolution images are essential for accurate quantification. Furthermore, the GFP channel need to be included to monitor correlation with GA aggregates. As the experiment was done by transient transfection, and it is well known that the transfection on neurons is not efficient, it is very likely that many neurons do not have GA aggregates, or express different levels of aggregates. In this case, why does the quantification show homogenous complete block of synaptic unloading? In addition, the statistics is not clear. "N=20 puncta regions" from how many neurons? Multiple neurons in each biological replicate should be included.
- Does the synaptic vesicle release stop completely or delayed? According to Fig 3B, it seems that the vesicle release stops within the 10s post-stimulation window. What is the relative fluorescence intensity after 10s till 2min (the end of the recording)?
- Is the degree of increase of Ca²⁺ influx upon stimulation correlated to the length of GA?
- Figure 5: again, the co-staining with GFP or GFP-GA is essential for the quantification, as it is not known which neurons are transfected. In Fig5.B, it seems SV2 in motor neurons forms bigger aggregates compared to GFP only group, instead of having a decreased number of puncta, and some do not look like puncta. Or the signal is over-saturated? The puncta size and intensity should be quantified. A western blot should be included to measure the total protein amount.
- More iPSN lines should be tested to demonstrate SV2 expression is down-regulated in C9 patients.
- The co-localization of GA and p62 is unconvincing. The image quality is low. The channels need to be separated in order to see the co-localization. Actually they look like mostly not co-localized. And it is not clear what's the relevance of this data to the rest of the manuscript.
- The longitudinal survival assay showed GA containing neurons display delayed toxicity. Would it be better to show this data after the GA mobility assay? Moreover, it is hard to understand Fig 7A and B (left panel). Should the numbers on the Y axis stand for the cumulative risk of death instead of cumulative survival? Any difference of the survival between GA50 and GA100? Why it seems that GA50 is more toxic than GA100? All the other assays showed that longer repeats have more severe phenotype than shorter repeats.
- The SV2 WB in Fig8A is not really convincing. The band of SV2 is not clear and there is huge variation among individuals within groups. It is hard to get the quantification and error bars on the right.
- For Fig9A, it would be better to replace NMJ with synaptophysin in the label. The left and right panel look very different. Does this mean even in wildtype animals, the expression of SV2 varies a lot? In this case, what is the quality control to make sure the same portion of NMJ was picked in GA149 mice? The quantification (intensity and area) of the SV2 signal in multiple NMJs from multiple animals is essential. It is also important to check the structure of NMJ with other markers, for example α -BTX.
- Representative images of Fig 9B are desired.

Minor points:

1. A thorough proofreading is required. Inconsistent writing has been noticed. For example, $P < 0.001$ is written as $P < .001$ sometimes (majorly in Fig5 and 6 associated paragraph and figure legend.).
2. In Fig.4, representative images of GFP control and no Ca²⁺ control need to be included.
3. In Fig7.A and B, the survival curve (blue line) is mis-labeled.
4. Fig EV1 and EV2 was placed in the wrong order in the merged file.

Referee #2 (Comments on Novelty/Model System for Author):

The findings are novel and potentially of importance to disease pathogenesis in C9orf72-mediated FTD/ALS. Previous studies (including those from this group) have testified to the toxicity of arginine-containing dipeptides, but this manuscript adds to the growing body of literature suggesting a separate and perhaps additive effect of GA.

The model systems (primary neurons, iPSC-derived neurons and transgenic mice) are excellent.

There is one concern regarding statistics in Fig. 7. For the survival analyses, they use the log-rank test, which demonstrates overall significance but not which specific comparisons are significant.

Referee #2 (Remarks for Author):

Here, Jensen et al. explore the impact mutant C9orf72-derived GA dipeptide repeats on synaptic function in neurons. They demonstrate that GA dipeptides localize to some degree in neuronal processes, and this phenomenon is associated with impaired synaptic vesicle release and a reduction in synaptic vesicle glycoprotein 2A (SV2). The findings are novel and potentially of importance to disease pathogenesis in C9orf72-mediated FTD/ALS. Previous studies (including those from this group) have testified to the toxicity of arginine-containing dipeptides, but this manuscript adds to the growing body of literature suggesting a separate and perhaps additive effect of GA.

The manuscript is well-written and the data are clearly presented. Even so, the primary observations (the presence of GA aggregates in neuronal processes, and reduced SV2 in GA-expressing neurons) remain largely disconnected with one another. Furthermore, there is little evidence indicating the relevance of SV2 to cellular toxicity and NMJ function in disease models. Addressing these issues would strengthen the conclusions and add to the overall significance of the findings.

Major Concerns:

- Data showing (a) GA in axons and dendrites, and (b) SV2 reductions in GA expressing cells are convincing, but as it stands there is no link between these observations.
- It remains unclear why the authors are seeing cell death in relation to GA overexpression in these cultures, but not in mice and not in their previous work (Wen et al 2014). Although the authors state that the toxicity is delayed beyond the time frame originally studied in Wen et al. (2014), the current experiment is only 4 days longer, and nearly all the difference occurs at between days 8-9. In fact, the relatively large jump in toxicity from day 8-9 suggests a rapid decline in cell health for unknown reasons.
- The contribution of reduced SV2 to the observed phenotypes is also unclear. Manipulating SV2 levels by knocking down in control neurons or overexpression in disease models may help resolve this question.

Specific concerns:

Figure 1:

- Quantification of percentage of cells positive for GA50 in dendrites and sample size would add to the impact of these observations. Also, it appears from the images in Fig. 1A that this phenomenon could be limited to cells with the highest expression level of the GA50 (i.e. the brightest cells).

- I also wonder whether the inclusions are specific to GA. Have the authors looked at other DPRs that form inclusions but are not associated with toxicity, such as GP or PA?

Figure 2: a good control for these studies would be GP or GA.

Figure 5: labeling of panels A and B with what has been stained for would improve clarity.

Figure 6:

- Would the authors expect SV2 to be reduced at the RNA as well as protein levels?
- The authors mention the heterogeneity of the motor neuron cultures as one reason for the lack of significant difference in SV2 levels by WB. This could be addressed by repeating the same analysis that they did in A and B of this figure (immunocytochemistry rather than WB).

Figure 7:

- It appears as if GA100 less toxic than GA50? The authors state that the log-rank test was used, but this demonstrates overall significance of the comparison, not which comparisons are significant.

- A good control would be GP or PA, or even GR/PR for relative toxicity.

- Also, why are the cells so much smaller in B? Are these motor neurons or spinal neurons, the latter of which contain a high proportion of interneurons?

- "GA" in the legend should be "eGFP".

Referee #3 (Comments on Novelty/Model System for Author):

The paper seems to me too observational to be meaningful in the context of decisively proving disease mechanism

Referee #3 (Remarks for Author):

The manuscript by Jensen et al. investigates the pathological effects caused by the dipeptide repeat proteins (DPRs), specifically by poly glysine alanine (GA) proteins, produced in the mouse GA model and neurons derived from patient iPSCs carrying C9orf72 hexanucleotide expansion. The author describe a set of phenotypes related to synaptic dysfunction, resulting from the loss of SV2.

There is a major concern that precludes the recommendation for publication. SV2 protein is implicated in many neuronal pathology-associated conditions, for example it is estimated to be reduced by 40% in epilepsy (SV2A is reduced by 40% compared to human brains lacking an epileptic focus (Feng et al., 2009)). How do the author explain the specificity of the ALS phenotype that results from the loss of SV2? Can it be a general long term consequence of neuronal pathology which manifests in distinct disorders, such as Alzheimer's disease, ALS, and epilepsy. If so how will the authors address this question.

Specific comments:

Figure 1 will benefit from showing GA channel separately from the Merged image. Additionally a quantification is necessary of the aggregate ratio on axon and dendrites.

The author compared GA aggregate and mitochondrial mobility, however this is not enough to conclude that the transport of other cargoes is not affected. If this conclusion is necessary for the work, other cargoes would have to be visualized.

2nd Revision - authors' response

6th Feb 2020

Referee #1. *Intronic G4C2 repeat expansion in C9orf72 is the most common cause of familial ALS/FTD. One of the pathogenic mechanism is the aberrant accumulation of five dipeptide repeat proteins produced by RAN translation. DPR inclusions have been detected in C9orf72 repeat expansion related ALS/FTD postmortem brain. Differ from the arginine-rich DPRs, namely GR and PR, GA aggregation is widely detected in brain and spinal cord causing more subtle cellular impairment, including neuronal loss, yet is still sufficient to trigger motor deficits. This phenomenon raises the possibility that GA expression might contribute to neuronal dysfunction without significant cell death. In the current paper, the authors utilized a serial of imaging methods showed that GA aggregates are mobile in the axon and dendrites of the neuron. Calcium influx and synaptic release are disrupted due to the presence of GA. Meanwhile, SV2 expression is down-regulated in multiple in vivo and in vitro models which the authors suggest contributing to the progression of the disease. While the paper raises an important question and provides some intriguing evidence that synaptic function is interrupted before neuronal death because of the presence of GA, there are some concerns and several data are not very convincing to support the conclusions. And there is no mechanism how GA aggregates lead to synapse dysfunction.*

Major concerns:

In Fig 1, the authors tried to show that GA50 aggregates are presented in the axon and dendrite of the primary cultured neuron. However, it is difficult to visualize the expression pattern of GA50 in neurons with the overexposed MAP2 and neurofilament signal. Especially the day 4 and day 8 images are not convincing. The GFP signal looks like background noise, as similar signals also shows in the extracellular space. A split channel of eGFP-GA50 and dimming neuronal marker signal is necessary to show the overall pattern of GA50 aggregates in the cell. Higher resolution images are necessary.

We have performed a secondary time course taking these considerations in mind so that the reader can better envision neuritic GA aggregates. Figure 1 now contains individual channels as well as a merge for each of the days (2,4,8).

How about the percentage of GA aggregates in soma vs neurite? Any difference between cortical neurons and motor neurons?

While we value this suggestion, each GA-containing cell has a highly variable number of aggregates. Many of which are mobile, as we have demonstrated in Figure 2. Furthermore, we have observed that the number of GA aggregates can vary in a cell and even dissipate over time, suggesting that the presence and localization of GA aggregates is a highly dynamic process. At any point in time that we have examined, some neurons (both cortical and motor) contain only perinuclear aggregates, a majority a mix of cytoplasmic and neuritic, and rare cells with only neuritic. We feel that quantifying this ever-fluctuating condition would not contribute further to the impact of this article. However, as suggested by reviewer 2, we have now included a quantification of the percentage of GA-aggregate containing cells which also contain neuritic aggregates, to solidify our stance that this is a major factor of GA-induced cellular effects. In addition, we also added in the text and discussion this very notion that the GA aggregates are very dynamic in nature and that certain cells have the ability to disaggregate them, as shown by the images that have now been added as **Extended View Figure 1B**.

An eGFP only transfection should be used as a control to show that the transfection itself does not contribute to the aggregation of the protein.

Assuming that the reviewer is referencing either Figure 1 or 2, GFP controls were performed in the cases of both of these experimental sets. We have now provided images for 8-day GFP transfected neurons to illustrate that GFP was not-aggregated at this time point in the absence of GA (**Extended View Figure 1A**). For Figure 2, GFP served as a transfection control, however since it was again cell filling, mobility was unable to be assessed so it was omitted from the graphs. This has now been made clearer in the text and the relevant figure legend.

In Fig.3A, it is hard to distinguish the real signal of FM4-64 dye from the background. What are the real signals at the synapse and what are extracellular signals? A bright field overlay or synapse labeling (such as by co-expressing with a synapse marker), as well as higher resolution images are essential for accurate quantification. Furthermore, the GFP channel needs to be included to monitor correlation with GA aggregates. As the experiment was done by transient transfection – and it is well known that the transfection of neurons is not efficient – it is very likely that many neurons do not have GA aggregates, or express different levels of aggregates. In this case, why does the quantification show homogenous complete block of synaptic unloading? In addition, the statistics is not clear. "N=20 puncta regions" from how many neurons? Multiple neurons in each biological replicate should be included.

We thank the review for these suggestions, and want to make clearer to the reviewers that only GFP⁺ neurons (i.e. neurons that ultimately express GA) were evaluated. Overall, we have greatly clarified the text description of how these experiments were performed. For each trial, the first step was to identify a GFP⁺ cell. A bright field, GFP, and TRITC image was taken for reference. Only synaptic puncta regions from that cell were monitored and quantified. Due to the resolution required and camera rate-capacity of 200ms, we are unable to take dual images of GA aggregates (green) while simultaneously monitoring synaptic release (red). We would love to be able to do so, but it is beyond the technological capabilities of our microscope cameras and – to the best of our knowledge – any cameras currently on the market. We have also better labeled our statistical description, so that it is clear that 5 puncta regions were selected from 4 independent biological replicates.

Does the synaptic vesicle release stop completely or delayed? According to Fig 3B, it seems that the vesicle release stops within the 10s post-stimulation window. What is the relative fluorescence intensity after 10s till 2min (the end of the recording)?

We have modified our stimulation parameters to better capture the full spectrum of vesicle release and have now also included SV2 upregulation conditions as discussed elsewhere. We have shortened our camera capture from 4-second intervals down to 2-second intervals, and now display the final 60 seconds of baseline and full period of recorded stimulation, as can be seen in **Figure 8**.

Is the degree of increase of Ca²⁺ influx upon stimulation correlated to the length of GA?

This was an excellent suggestion, and one that we had considered pursuing. We performed length-curve experiments to determine if length of GA peptide would alter the degree of calcium influx, now included as part of **Extended View Figure 3**. There is not a length-dependent effect

observed, rather the presence of GA aggregates of any length was sufficient to cause increase in calcium influx to roughly the same extent in all the GA lengths tested. A comment on this was added in the manuscript.

Figure 5: again, the co-staining with GFP or GFP-GA is essential for the quantification, as it is not known which neurons are transfected. In Fig5.B, it seems SV2 in motor neurons forms bigger aggregates compared to GFP only group, instead of having a decreased number of puncta, and some do not look like puncta. Or the signal is over-saturated? The puncta size and intensity should be quantified. A western blot should be included to measure the total protein amount.

We have attempted to address this concern in multiple ways. We have clarified our methodological explanation to clarify that SV2 levels were evaluated only in GFP+ cells. We have performed additional high-resolution imaging to present a more accurate representation of aggregates in healthy neurites of transfected motor neurons. Criteria for inclusion of a “puncta” is now included in the materials and methods description of this quantification. Furthermore, we have performed high resolution imaging to determine SV2 and GA aggregate colocalization (**Extended View 4**). However, we feel that we will not be able to perform a rigorous, meaningful and reproducible GA aggregate size profile even if we were to use a more advanced confocal or STED microscopy system than we presently have access to, due to the highly dynamic nature of these very same aggregates. Since these aggregates are not at equilibrium, we feel that the degree of variability of this kind of size profiling won't be reproducible and therefore informative. In addition, due to the transfection efficiency in primary neuronal culture, performing a Western Blot to evaluate SV2 levels is unfeasible in this system. However, it is our intention to address this point during experiments designed to determine further mechanism underlying GA-induced dysregulation of SV2. We have recently procured GA lentivirus to enable high efficiency transduction and GA expression. A Western Blot of these cultures, as well as analysis by cryo-EM will be forthcoming in a follow-up study.

More iPSN lines should be tested to demonstrate SV2 expression is down-regulated in C9 patients.

We realize that our number of i3 neuron lines is a limitation of our iPS experiments. At present no additional lines exist in this format, though we are in the process of generating new i3 lines from C9 patient fibroblasts as well as paired CRISPR-edited isogenics. This is why we also included data from the 4 C9 and 5 control iMN lines that we had access to. Additionally, in an attempt to bolster our claim from the one paired same that we do have, in addition to immunofluorescent staining and RNA quantification, we have now also included an evaluation of SV2 protein levels from the i3 neurons and paired isogenic control (**Figure 7F**).

The co-localization of GA and p62 is unconvincing. The image quality is low. The channels need to be separated in order to see the co-localization. Actually, they look like mostly not co-localized. And it is not clear what's the relevance of this data to the rest of the manuscript.

We appreciate the reviewer bringing up this point. We feel that we were unclear in how we described this figure, and did not want to make any claims about amount of GA colocalized with p62, simply that some did. This was included as confirmatory staining of previously published work, but we agree that it does not substantially contribute to the present manuscript. It has now been omitted.

The longitudinal survival assay showed GA containing neurons display delayed toxicity. Would it be better to show this data after the GA mobility assay? Moreover, it is hard to understand Fig 7A and B (left panel). Should the numbers on the Y axis stand for the cumulative risk of death instead of cumulative survival? Any difference of the survival between GA50 and GA100? Why it seems that GA50 is more toxic than GA100? All the other assays showed that longer repeats have more severe phenotype than shorter repeats.

We have given additional consideration to the placement of this Figure in light of the reviewers comment. We have decided to follow this plan of rearrangement, and the Longitudinal survival assessment of GA containing cells is now presented as **Figure 3** instead of Figure 7. The reviewer is correct that we have mislabeled the Y-axis, which should have been cumulative risk of death. We have subsequently performed several “Rescue” experiments in which we upregulated SV2 levels by lentiviral transduction. As such, we have now included additional data within these Kaplan-Meier assessments. However, this unusual finding of heightened GA50 toxicity compared with GA100 persists. We feel that since GA50 is highly mobile as well, it may have increased

opportunities to interact with other cellular players so as to heighten toxic effects. This point has now been addressed in the corresponding Results and discussion sections.

The SV2 WB in Fig8A is not really convincing. The band of SV2 is not clear and there is huge variation among individuals within groups. It is hard to get the quantification and error bars on the right.

We value the feedback regarding these blots. While inter-animal variation does still exist, we have generated Western Blots that more accurately reflect the densitometric quantification presented in Figure 8 (now **Figure 9**).

For Fig9A, it would be better to replace NMJ with synaptophysin in the label. The left and right panel look very different. Does this mean even in wildtype animals, the expression of SV2 varies a lot? In this case, what is the quality control to make sure the same portion of NMJ was picked in GA149 mice? The quantification (intensity and area) of the SV2 signal in multiple NMJs from multiple animals is essential. It is also important to check the structure of NMJ with other markers, for example α -BTX.

Representative images of Fig 9B are desired.

The staining as indicated “NMJ” within figure 9A (now 10A) was performed with α -BTX, and the figure has been modified to reflect this. Overall for this figure, we have improved the quality of the representative images provided, and added representative neurofilament co-staining with SV2 as recommended for Figure 1B. As the reviewer observed, there is some variability in SV2 coverage for each NMJ, and as such we have been more explicit in adding our replicates for this experiment into the methods section, rather than just in the figure legend. To minimize this intra-animal variation, 10 non-overlapping regions of muscle were evaluated in 4-5 animals per experimental group.

Minor points:

1. A thorough proofreading is required. Inconsistent writing has been noticed.

For example, $P < 0.001$ is written as $P < .001$ sometimes (majorly in Fig5 and 6 associated paragraph and figure legend.).

We thank the reviewer for this comment, and have performed a thorough proofread and suggested.

2. In Fig.4, representative images of GFP control and no Ca^{2+} control need to be included.

We thank the reviewer for this suggestion. We have restructured this figure to include informative representative images, as well as adding in a GA length curve as recommended in Expanded View Figure 3A.

3. In Fig7.A and B, the survival curve (blue line) is mis-labeled.

We appreciate this feedback. We have corrected this error and have greatly expanded upon this figure as is noted in our introductory paragraph regarding SV2-pRRL upregulation rescue experiments.

4. Fig EV1 and EV2 was placed in the wrong order in the merged file.

We apologize for this oversight and have ensured the files are in the correct order going forward.

Referee #2 (Comments on Novelty/Model System for Author): *The findings are novel and potentially of importance to disease pathogenesis in C9orf72-mediated FTD/ALS. Previous studies (including those from this group) have testified to the toxicity of arginine-containing dipeptides, but this manuscript adds to the growing body of literature suggesting a separate and perhaps additive effect of GA. The model systems (primary neurons, iPSC-derived neurons and transgenic mice) are excellent. There is one concern regarding statistics in Fig. 7. For the survival analyses, they use the log-rank test, which demonstrates overall significance but not which specific comparisons are significant.*

Referee #2 (Remarks for Author): *Here, Jensen et al. explore the impact mutant C9orf72-derived GA dipeptide repeats on synaptic function in neurons. They demonstrate that GA dipeptides localize to some degree in neuronal processes, and this phenomenon is associated with impaired synaptic vesicle release and a reduction in synaptic vesicle glycoprotein 2A (SV2). The findings are novel and potentially of importance to disease pathogenesis in C9orf72-mediated FTD/ALS. Previous studies (including those from this group) have testified to the toxicity of arginine-containing dipeptides, but this manuscript adds to the growing body of literature suggesting a separate and perhaps additive effect of GA.*

The manuscript is well-written and the data are clearly presented. Even so, the primary observations (the presence of GA aggregates in neuronal processes, and reduced SV2 in GA-expressing neurons) remain largely disconnected with one another. Furthermore, there is little evidence indicating the relevance of SV2 to cellular toxicity and NMJ function in disease models. Addressing these issues would strengthen the conclusions and add to the overall significance of the findings.

Major Concerns:

Data showing (a) GA in axons and dendrites, and (b) SV2 reductions in GA expressing cells are convincing, but as it stands there is no link between these observations.

We understand that this was a major concern of all three Reviewers upon our initial submission. Please see our introductory paragraph for full details on how we have probed for a mechanistic link between the presence of GA and SV2 downregulation in this resubmission.

It remains unclear why the authors are seeing cell death in relation to GA overexpression in these cultures, but not in mice and not in their previous work (Wen et al 2014). Although the authors state that the toxicity is delayed beyond the time frame originally studied in Wen et al. (2014), the current experiment is only 4 days longer, and nearly all the difference occurs at between days 8-9. In fact, the relatively large jump in toxicity from day 8-9 suggests a rapid decline in cell health for unknown reasons.

The reviewer brings up a valid point, that overall the field is undecided regarding the toxic effects of GA dipeptides. Within this revision, we have provided additional experiments to suggest that downregulation of SV2 in these enriched neuronal cultures seems to be a trigger for hastened cell death. Upregulation of SV2 via lentiviral transduction was sufficient to significantly lessen toxicity produced by each GA dipeptide length. Furthermore, the rapid decline between days 8-9 has resolved as an artifact related to the media change which occurs at that time period. We feel that in the isolated *in vitro* neuronal environment GA is sufficient to induce a slowly progressing cellular toxicity, which becomes apparent towards the end of our normal culturing period (21 days for cortical and 14 days for motor neurons). In the *in vivo* setting, we suspect that the more comprehensive cellular environment of neurons surrounded by glia enables neurons to survive even with GA aggregates and lower levels of SV2 for longer periods of time. This is consistent with patient post-mortem tissue bearing GA aggregates in what appear to be surviving motor and cortical neurons.

The contribution of reduced SV2 to the observed phenotypes is also unclear. Manipulating SV2 levels by knocking down in control neurons or overexpression in disease models may help resolve this question.

We understand that this was a high priority concern upon our initial submission. Please see our introductory paragraph for full details on how we have attempted to probe the contribution of SV2 to the phenotypes described in this resubmission.

Specific concerns:

Figure 1: Quantification of percentage of cells positive for GA50 in dendrites and sample size would add to the impact of these observations. Also, it appears from the images in Fig. 1A that this phenomenon could be limited to cells with the highest expression level of the GA50 (i.e. the brightest cells).

See response to point 2 of Reviewer 1.

I also wonder whether the inclusions are specific to GA. Have the authors looked at other DPRs that form inclusions but are not associated with toxicity, such as GP or PA?

We have considered this in the past and published our findings as supplemental information in our original paper by Wen *et al.* among the DPRs, only GA formed neuritic inclusions. A reference in the main text of this manuscript has now been made to further clarify the uniqueness of GA localization pattern.

Figure 2: a good control for these studies would be GP or GA.

In our hands, when transfected in cortical or motor neurons GP and PA are primarily diffuse, and what aggregates do form remain in the cell body as indicated in our response above. To hopefully address your point of controls for trafficking within the neurites, in addition to our mitochondrial data, we have now added additional assays probing the mobility of lysosomes and total RNA, which each contain mCherry transfected neurons as a negative control.

Figure 5: labeling of panels A and B with what has been stained for would improve clarity.

We thank the reviewer for this suggestion. We have incorporated this change as well as making the additional changes to original Figure 5 as requested by Reviewer 1.

Figure 6: Would the authors expect SV2 to be reduced at the RNA as well as protein levels?

In an attempt to describe a mechanism of action, we have now expanded our initial results to include an evaluation of SV2 RNA levels in GA-containing cells, to determine colocalization of GA with SV2 RNA. See introductory paragraph for full details.

The authors mention the heterogeneity of the motor neuron cultures as one reason for the lack of significant difference in SV2 levels by WB. This could be addressed by repeating the same analysis that they did in A and B of this figure (immunocytochemistry rather than WB).

Unfortunately, we were limited to cellular lysates from the iMN cultures provided by Answer ALS. However, to substantiate our iPS data we have now included an evaluation of SV2 protein in our isneuronal cultures, which backs up our immunofluorescent staining in that paradigm (**Figure 7**).

Figure 7: It appears as if GA100 less toxic than GA50? The authors state that the log-rank test was used, but this demonstrates overall significance of the comparison, not which comparisons are significant. A good control would be GP or PA, or even GR/PR for relative toxicity. Also, why are the cells so much smaller in B? Are these motor neurons or spinal neurons, the latter of which contain a high proportion of interneurons? "GA" in the legend should be "eGFP".

We have addressed some of the reviewer's concerns regarding the longitudinal survival assay in the previous response under major concerns. We have also addressed the concern regarding GA50 versus GA100 toxicity in response to a similar point made by Reviewer 1. In restructuring this figure, we have now included individual Hazard Ratio values for each experimental group in addition to providing the log-rank test results. In preparing our revised figure including SV2-pRRL Rescue conditions, we have also included PR as normalizing control for toxicity. The author is correct that we had an error in scale bar labeling in regards to our original panels. The cortical neuron images had been enlarged so that the reader could appreciate the dendritic complexity, and this scale inadvertently carried over to the motor neuron images. We have ensured correct labeling within the text and figure legend that we are working with primary spinal motor neurons.

Referee #3 (Comments on Novelty/Model System for Author): *The paper seems to me too observational to be meaningful in the context of decisively proving disease mechanism*

Referee #3 (Remarks for Author): *The manuscript by Jensen et al. investigates the pathological effects caused by the dipeptide repeat proteins (DPRs), specifically by poly glysine alanine (GA) proteins, produced in the mouse GA model and neurons derived from patient iPSCs carrying C9orf72 hexanucleotide expansion. The authors describe a set of phenotypes related to synaptic dysfunction, resulting from the loss of SV2. There is a major concern that precludes the recommendation for publication. SV2 protein is implicated in many neuronal pathology-associated conditions, for example it is estimated to be reduced by 40% in epilepsy (SV2A is reduced by 40% compared to human brains lacking an epileptic focus (Feng et al., 2009)). How do the author explain the specificity of the ALS phenotype that results from the loss of SV2? Can it be a general long term consequence of neuronal pathology which manifests in distinct disorders, such as Alzheimer's disease, ALS, and epilepsy. If so how will the authors address this question.*

The authors would like to address this issue by remarking that they do not intend to imply that the SV2 downregulation that they observe in the presence of GA-dipeptides is the entire cause of the ALS phenotype in patients, but is an important contributing factor preventive effective synaptic communication of motor neurons with each other and with muscle. In the context of C9orf72 it is abundantly clear that the other dipeptide repeat proteins and RNA foci also produce independent toxicities. It is likely that all 3 of the proposed mechanisms for C9orf72 (dipeptide repeat toxicity, RNA toxicity, and C9orf72 haploinsufficiency) act combinatorially in patients to cause disease. We have now framed our introduction and discussion to reflect these points, and suggest that SV2 may be a potential therapeutic target for the GA-mediated effects in patients, but that it would not be a “cure-all” for C9 patients. The importance of addressing GA dipeptides as a therapeutic target was also demonstrated by a work recently published by two independent groups: One in Neuron by the Ranum's group (PMID:31831332 *Antibody Therapy Targeting RAN Proteins Rescues C9 ALS/FTD Phenotypes in C9orf72 Mouse Model*), which has been added to our introduction as well as discussion; The second one in this very journal, published just recently by the Edbauer's group (*Active poly-GA vaccination prevents microglia activation and motor deficits in a C9orf72 mouse model*).

Specific comments: Figure 1 will benefit from showing GA channel separately from the Merged image. Additionally, a quantification is necessary of the aggregate ration on axon and dendrites.

See response to points 1 and 2 from Reviewer 1.

The author compared GA aggregate and mitochondrial mobility, however this is not enough to conclude that the transport of other cargoes is not affected. If this conclusion is necessary for the work, other cargoes would have to be visualized.

We agree with this assessment and have now included further trafficking assays. In addition to mitochondrial mobility, we now present data probing the mobility of lysosomes as well as total RNA (**Extended View Figure 2**).

3rd Editorial Decision

9th Mar 2020

Thank you for the submission of your revised manuscript to EMBO Molecular Medicine. We have now received the enclosed reports from the three referees that were asked to re-assess it. As you will see the reviewers are now globally supportive and I am pleased to inform you that we will be able to accept your manuscript pending the following final amendments:

1) Please address the minor text change commented by referee 2. In addition, regarding the rescue experiment comment, should you have data at hand or could replace the Western Blot with a more convincing one, we'd be happy for you to include it. Please address the referees' comments in writing and provide a point-by-point letter INCLUDING my comments as well as the reviewer's reports and your detailed responses to their comments (as Word file).

***** Reviewer's comments *****

Referee #1:

[This referee is satisfied with the revision and has no further comment]

Referee #2 (Comments on Novelty/Model System for Author):

The revision is significantly improved over the initial submission, but critical questions or concerns were not addressed.

Referee #2 (Remarks for Author):

The revision is significantly improved over the initial submission, but critical questions or concerns were not addressed.

Original Concern: Data showing (a) GA in axons and dendrites, and (b) SV2 reductions in GA expressing cells are convincing, but as it stands there is no link between these observations.

Author response: We understand that this was a major concern of all three Reviewers upon our initial submission. Please see our introductory paragraph for full details on how we have probed for a mechanistic link between the presence of GA and SV2 downregulation in this resubmission.

→ It is unclear precisely what introductory paragraph the authors referring to. If they are referring to the bottom of page 4, this short addition does not describe a mechanistic link between GA and SV2. There is no indication of how GA expression may lead to SV2 reduction, for instance. Further, while it is certainly interesting and significant that SV2 upregulation rescues some of the toxicity from GA expression, it also seems in the new Figure 8 that SV2 overexpression improves overall survival in control neurons, suggesting a more general effect.

Concern: It remains unclear why the authors are seeing cell death in relation to GA overexpression in these cultures, but not in mice and not in their previous work (Wen et al 2014). Although the authors state that the toxicity is delayed beyond the time frame originally studied in Wen et al. (2014), the current experiment is only 4 days longer, and nearly all the difference occurs at between days 8-9. In fact, the relatively large jump in toxicity from day 8-9 suggests a rapid decline in cell health for unknown reasons.

Author response: The reviewer brings up a valid point, that overall the field is undecided regarding the toxic effects of GA dipeptides. Within this revision, we have provided additional experiments to suggest that downregulation of SV2 in these enriched neuronal cultures seems to be a trigger for hastened cell death. Upregulation of SV2 via lentiviral transduction was sufficient to significantly lessen toxicity produced by each GA dipeptide length. Furthermore, the rapid decline between days 8-9 has resolved as an artifact related to the media change which occurs at that time period. We feel that in the isolated in vitro neuronal environment GA is sufficient to induce a slowly progressing cellular toxicity, which becomes apparent towards the end of our normal culturing period (21 days for cortical and 14 days for motor neurons). In the in vivo setting, we suspect that the more comprehensive cellular environment of neurons surrounded by glia enables neurons to survive even with GA aggregates and lower levels of SV2 for longer periods of time. This is consistent with patient post-mortem tissue bearing GA aggregates in what appear to be surviving motor and cortical neurons.

→ This revision appears to include completely different data on the survival of GA-expressing neurons. The new data are more consistent with a moderate effect of GA on the survival of primary neurons. Even so, in comparing the new Fig 3 and Fig 8 with Fig 1B from Wen et al. (2014), GA50 appears to be more toxic by day 8 in the newer experiments, for unclear reasons. In fact, GA50 (in Fig. 8C) is just as toxic as PR50 (in Fig. 8D), but in Fig. 1B from Wen et al. (2014) PR50 is substantially more toxic than any other DPR, including GA. This difference should be addressed or commented on by the authors, since readers familiar with their work may be wondering at the newer data and why a similar effect was not observed earlier.

Also, in the new Fig 8, panels C and E are almost identical, with the exception of the control + SV2 condition. C could very well be replaced with E.

The SV2 rescue experiment is an important one, and a strong addition to the manuscript. To confirm this rescue, it would be helpful if the authors can show (by ICC perhaps) if the cells that they are following in Fig. 8 C-E do indeed express higher amounts of SV2 protein, since there is no indication of transduction efficiency and the WB result demonstrating protein overexpression with transduction is borderline (Fig. EV4).

Concern: Figure 2: a good control for these studies would be GP or GA.

Author response: In our hands, when transfected in cortical or motor neurons GP and PA are primarily diffuse, and what aggregates do form remain in the cell body as indicated in our response above. To hopefully address your point of controls for trafficking within the neurites, in addition to our mitochondrial data, we have now added additional assays probing the mobility of lysosomes and total RNA, which each contain mCherry transfected neurons as a negative control.

→ The assessment of different phenotypes is a good addition, but does not address the question of whether other DPRs may cause similar effects. This becomes even more important after the addition of new data in Fig. 8 showing that SV2 expression rescues GA- but not PR-induced toxicity.

Referee #3 (Remarks for Author):

I think that the revision is satisfactory and that it is possible to proceed to publication.

3rd Revision - authors' response

23rd Mar 2020

Referee #1: This referee is satisfied with the revision and has no further comment

Referee #2:

→ It is unclear precisely what introductory paragraph the authors referring to. If they are referring to the bottom of page 4, this short addition does not describe a mechanistic link between GA and SV2. There is no indication of how GA expression may lead to SV2 reduction, for instance. Further, while it is certainly interesting and significant that SV2 upregulation rescues some of the toxicity from GA expression, it also seems in the new Figure 8 that SV2 overexpression improves overall survival in control neurons, suggesting a more general effect.

We regret the confusion of which introductory paragraph we were referring to in our previous response. We intended this to be read as the introduction to the rebuttal letter, in which we addressed our initial strategy to determine a mechanistic link between GA and SV2, as well as providing a compelling therapeutic target through SV2 upregulation as suggested. This paragraph is written again here below: “The most pressing issue, brought up by all three reviewers, was a disconnect between our finding of GA aggregates in neurites with no mechanistic link tying this localization to our observed downregulation of SV2 levels or how this would cause cellular dysfunction. To respond to this shared concern, we have done experiments directly linking GA aggregates with both SV2 protein and RNA, which showed that the presence of GA inclusions prevent SV2 appropriate production and localization to synaptic vesicles. In addition to high-resolution colocalization of GA aggregates and SV2 protein (Figure 8A-B), we have included SV2-RNA beacon experiments showing colocalization of GA aggregates with SV2 mRNA (Figure 8C-D). Furthermore, to determine if SV2 levels were the root cause for the other cellular dysfunctions that we observed, SV2 upregulation “rescue” experiments were performed using SV2 Rpl lentivirus that was kindly shared with us from the Bajalieh laboratory. Following SV2 upregulation, we found that calcium influx levels were normalized, synaptic transmission was restored, and cellular toxicity was attenuated (Figure 8).” Furthermore in addressing the final point of this concern, we only saw an overall survival improvement upon SV2 upregulation in control motor neurons (Fig 8E). When examined in cortical neurons (Fig 8C), upregulation of SV2 in control neurons did not alter the survival curve, which is commented on in within the figure

legend. All of our functional tests were performed in the cortical neuron setting, suggesting that restoration of SV2 levels did contribute to the mitigation of GA-induced synaptic deficits. The enhancement of survival in control motor neurons is certainly intriguing, and is worth pursuing even in the absence of ALS phenotypes in future studies.

→ This revision appears to include completely different data on the survival of GA-expressing neurons. The new data are more consistent with a moderate effect of GA on the survival of primary neurons. Even so, in comparing the new Fig 3 and Fig 8 with Fig 1B from Wen et al. (2014), GA50 appears to be more toxic by day 8 in the newer experiments, for unclear reasons. In fact, GA50 (in Fig. 8C) is just as toxic as PR50 (in Fig. 8D), but in Fig. 1B from Wen et al. (2014) PR50 is substantially more toxic than any other DPR, including GA. This difference should be addressed or commented on by the authors, since readers familiar with their work may be wondering at the newer data and why a similar effect was not observed earlier.

We acknowledge that there are greatly increased numbers of cells included in our presented survival analysis, which has resolved the original concerns. This is not completely different data, the original data set is still included, now along with an additional 3 replicates. There are a number of reasons which we believe have contributed to why our present survival data differs from our original curves presented in Wen et al. 2014. In our original paper, we probed for overall effects of all DPRs using 90,000 cortical neurons per well on PDL and laminin-coated plates treated with AraC. These cells were transfected on DIV10. In these newer experiments optimized for GA expression, 125,000 cortical neurons per well were plated on PDL coated plates and transfected on DIV7. No laminin or AraC were used. There are differences in the survival curves of the control neurons with these alterations, suggesting differences in cellular sensitivity when transfected on different days in vitro. In particular, the immediate and dramatic toxicity associated with PR in our original study may be due in part to the later DIV at which these cells were transfected, as well as the harsher environment caused by the AraC treatment. Since this time, many others have also confirmed the toxicity of the PR and GR species under various culture conditions. The GA curves now presented were begun at a slightly earlier timepoint to allow for an extended period of evaluation. The bottom line is, however, that what was originally reported as a trend toward toxicity (Fig.1B in Wen et al.) is now confirmed as being statistically significant with the optimized conditions.

Also, in the new Fig 8, panels C and E are almost identical, with the exception of the control + SV2 condition. C could very well be replaced with E.

We agree that panels 8C and 8E do look similar. However, as is noted in the text Figure 8C refers to cortical neurons, while Figure 8E refers to motor neurons. Therefore, they cannot be combined into a single graph.

The SV2 rescue experiment is an important one, and a strong addition to the manuscript. To confirm this rescue, it would be helpful if the authors can show (by ICC perhaps) if the cells that they are following in Fig. 8 C-E do indeed express higher amounts of SV2 protein, since there is no indication of transduction efficiency and the WB result demonstrating protein overexpression with transduction is borderline (Fig. EV4).

We appreciate this feedback. We have performed additional experiments to address this concern and have now included the data in Figure EV4. An additional replicate of western blotting for SV2 better illustrates the quantified ~1.5 fold induction of SV2 at the protein level in cortical neurons (Fig. EV4 D). Furthermore, as recommended, immunofluorescent staining showed an enhancement of SV2 puncta in control neurons, and a rescue of SV2 puncta levels back to control levels in GA-expressing neurons (Fig. EV4 E-F).

→ The assessment of different phenotypes is a good addition, but does not address the question of whether other DPRs may cause similar effects. This becomes even more important after the addition of new data in Fig. 8 showing that SV2 expression rescues GA- but not PR-induced toxicity.

We agree that understanding whether each of the DPRs produces synaptic effects is an interesting scientific inquiry. As we mentioned in our previous response, within our original paper by Wen et al. we noted that among the DPRs, only GA formed neuritic inclusions and we therefore suspected only

GA would have specific synaptic effects. At that time we did perform synaptic release experiments to probe whether any of the DPRs have deficits following 2 days of expression in motor neurons. Significant reduction in unloading was seen with the robustly toxic GR and PR dipeptides, and with GA. Neither GP nor PA showed reduction compared with the GFP control (the unpublished data shown in the figure below is only for reviewer evaluation and has not been included in the manuscript main figures nor as an EV figure).

Synaptic vesicle unloading of cortical neuron neurites induced by KCl depolarization is compromised by GA₅₀, PR₅₀ and GR₅₀ dipeptides, while GP₅₀ and PA₅₀ dipeptides are ineffective. Motor neurons were loaded with FM4-64 at 48 hours post transfection. Prior to dye loading, cells were incubated in Ca₂₊-free ACSF for 10 minutes at 37 °C. Neurons were then incubated in the dark at room temperature with 10 μM FM4-64 in high KCl for 5 minutes followed by perfusion with Ca₂₊-free ACSF for 10 minutes to eliminate non-specific dye loading. Baseline fluorescence was recorded to exclude

variations in dye intensity. KCl (50 mM) was then constantly perfused to facilitate unloading of FM4-64. Bar graphs show the average loss of FM4-64 fluorescence of at least 3 motor neurons/group in 3 different experiments. Statistical significance compared to the GFP group is indicated where appropriate.

Rather than isolating single DPRs, future experiments will aim to explore synaptic function in patient derived C9or72 neurons, to evaluate the more complex patient-environment with the full complement of DPRs and RNA species.

Referee #3: I think that the revision is satisfactory and that it is possible to proceed to publication.

4th Editorial Decision

26th Mar 2020

Thank you for submitting your revised article. We are pleased to inform you that your manuscript is accepted for publication.

Corresponding Author Name: Liangyi Chen and Guangshuo Ou

Journal Submitted to: EMBO Journal Research Articles

Manuscript Number: EMBOJ-2019-103955R